# Thermal State of Permafrost in the Central Andes (27°S-34°S)

Cassandra E.M. Koenig[1,2], Christin Hilbich[1], Christian Hauck[1], Lukas U. Arenson[3] & Pablo Wainstein[4]

[1]Department of Geosciences, University of Fribourg, Fribourg, Switzerland
[2]BGC Engineering Inc., Toronto, ON, Canada
[3]BGC Engineering Inc., Vancouver, BC, Canada
[4]BGC Engineering Inc., Calgary, AB, Canada

*Correspondence to*: Cassandra Koenig (cassandra.koenig@unifr.ch)

## Abstract

Permafrost thermal state is poorly understood in South America compared to other regions due to limited in-situ data. This study represents the first coordinated regional compilation of borehole temperature data from high-altitude mountain permafrost sites in the Central Andes (3,500 m to 5,250 m, 27°S-34°S) to examine baseline ground thermal conditions. Measurements from 53 boreholes along a north-south transect at the Chilean-Argentine border reveal ground thermal characteristics similar to other mountain permafrost regions, including high spatial and temporal variability, correlations with altitude and slope aspect, and distinct thermal attributes of rock glaciers. Observations suggest that the ground thermal regime of the Central Andes is shaped by similar processes as other mountain environments, a perspective that was previously lacking data support. The high temporal variability observed in the short records (<9 years) reflects short-term microclimatic fluctuations and topo-climatic attributes unique to the Andean cryosphere. This includes hyper-arid conditions, intense solar radiation, limited vegetative cover and organic matter, less massive ice (except for rock glaciers), and mountain topography in a southern hemisphere location. The susceptibility of the area to regional climatic phenomena (such as SAM, ENSO and PDO) suggests that long-term trends may only be determined from extended datasets spanning several decades. This study highlights the need for ongoing ground temperature monitoring, and the importance of collaboration between industry, governments, and scientists to advance understanding of a key climate change indicator in a data-scarce region.

## 1 Introduction

Permafrost is an Essential Climate Variable (ECV) that serves as a cryospheric indicator of climate change (WMO, 2016; Streletskiy et al., 2017). Several studies have documented large-scale warming and thawing of permafrost in recent decades (e.g., Romanovsky et al., 2017; Derksen et al., 2019; Etzelmüller et al., 2020; Haberkorn et al., 2021; Nyland et al., 2021). Current research suggests that permafrost will continue to warm in many regions in the near term (i.e., 2031-2050) due to projected global increases in surface air temperatures, with acceleration expected in the second half of the 21st century under extreme shared socioeconomic pathway scenarios (Abram et al., 2019).

Baseline permafrost monitoring data is essential for risk-informed engineering design and environmental impact assessments of resource development projects. In mountain regions, near-surface (i.e., 0-10 m) thawing of permafrost significantly affects land stability, causing subsidence, slope failures, and water release from ice-rich landforms (Gruber and Haeberli, 2007; Krautblatter et al., 2010; Romanovsky et al., 2017; Kokelj et al., 2017). In Arctic and boreal regions, where permafrost soils store significant amounts of organic carbon (e.g. Schuur et al., 2009), permafrost thaw may amplify surface warming via the

permafrost carbon-climate feedback (Schaefer et al., 2014; Koven et al., 2015; Schuur et al., 2015; Miner et al., 2022; Schuur et al., 2022). Understanding the present and potential future thermal conditions of permafrost is therefore essential for guiding robust infrastructure development and mitigating environmental risks in the face of climate change.

    Two important parameters for characterizing and monitoring permafrost thermal state are 1) active layer thickness (ALT), or depth to permafrost; and 2) ground temperature measured at or below the depth of zero annual amplitude (DZAA). The active

layer, classically defined as the layer above permafrost that freezes and thaws annually (van Everdingen, 1998), delineates the depth to the top of permafrost when measured at the time of maximum annual thaw. This is not necessarily the same as the depth to the permafrost table particularly where permafrost is degrading and a talik has formed. The DZAA represents the depth at which seasonal temperature variations are fully attenuated by the ground, often defined as the depth where amplitudes diminish to 0.1°C or less (van Everdingen, 1998). Ground temperature at or below the DZAA is therefore considered a good

indicator of long-term permafrost thermal state because it represents the mean annual temperature of the ground over time (Lachenbruch and Marshall, 1986). Active layer thickness and temperature at the DZAA are both affected by climate change, particularly rising air temperatures, changes to precipitation patterns and snow distribution, which influence heat transfer from the ground surface to the subsurface (Isaksen et al., 2007; Christiansen et al., 2010; Biskaborn et al., 2019).

    Most sites gathering standardized permafrost temperature data are in polar and high mountain areas of North America, Europe

and Russia/Siberia, with fewer sites located in Central Asia, Antarctica, and South America (e.g., Biskaborn et al., 2015; Brown et al., 2000; PERMOS, 2019). At some locations, permafrost thermal state has been documented continuously for up to 40 years (e.g., Romanovsky, et al., 2010b), showing increasing permafrost temperatures, particularly in the Arctic (Romanovsky et al., 2010a). However, assessing long-term changes in response to climate change remains challenging at many locations due to sparse borehole distribution and short or discontinuous monitoring records. These challenges often arise from

site access difficulties, high installation and maintenance costs and the narrow focus of monitoring programs, particularly in mountain permafrost regions (Biskaborn et al., 2019; Noetzli et al., 2021; Smith et al., 2022).

    The lack of ground temperature data in permafrost research, especially at depths beyond a few metres, is particularly evident in South America, where permafrost is largely understudied compared to other regions of the world (Arenson et al., 2022). Despite a long-standing awareness of permafrost in the Andes (e.g. Catalano, 1926; Corte 1953), ground-based studies remain

limited due to the region's high elevations, harsh climate and rugged terrain. Challenges such as limited funding and inadequate infrastructure for accessing remote locations further complicate data acquisition (Arenson and Jakob, 2010; Hilbich et al., 2022; Mathys et al., 2022). While some ground temperature monitoring studies in the Andes have been published (e.g., Trombotto et al., 1997; Trombotto and Borzotta, 2009; DGA, 2010; Andrés et al., 2011; Monnier and Kinnard, 2013; Atacama

Ambiente, 2017; DGA, 2019; Nagy et al., 2019; Yoshikawa et al., 2020; Mena et al., 2021; Vivero et al., 2021; Table 1), most instruments have not been established for the explicit purpose of monitoring permafrost thermal change with time. Consequently, monitoring records often lack the duration and depth required to discern average ground temperatures or trends in warming/cooling below the DZAA, limiting characterization of the ground thermal state in the Andes compared to regions where large-scale permafrost degradation has been extensively documented. Many ground temperature monitoring studies in South America have focused on estimating the lower regional altitude of permafrost, collecting measurements only up to one metre deep and over periods shorter than five years (Andrés et al., 2011; Nagy et al., 2019; Yoshikawa et al., 2020; Mena et al., 2021; Vivero et al., 2021). This lack of deeper measurements has been acknowledged in a proposed permafrost national monitoring plan for Chile (DGA, 2019), which recommends long-term monitoring using boreholes extending to the base of permafrost. However, no installations under this plan have exceeded 2 m in depth to date (Table 1).

Some permafrost studies in the Andes have utilized deeper boreholes to characterize permafrost thermal conditions and to monitor changes in temperature over time. Trombotto and Borzotta (2009) documented permafrost degradation in a rock glacier in Argentina in a 5-m-deep borehole over nearly 10 years, with annual increases in ALT by up to 25 cm. Monnier and Kinnard (2013) reported findings from two boreholes with thermistor strings reaching 18-25 m in the upper Choapa valley of northern Chile. Monitoring of the 25-m-deep borehole between 2010 and 2013 showed stable temperatures near 0°C along the profile, with an active layer estimated to range between 5-7 m thick (Monnier and Kinnard, 2013). Preliminary data from three boreholes (20-40 m deep) installed at the Goldfields Salares Norte mining project in Chile, indicated favorable conditions for the presence of permafrost between approximately 5 m and 13 m depth at one borehole (Atacama Ambiente, 2017). Although these studies provide valuable insights into the thermal state and changes to permafrost, the fact that they represent the bulk of time-series measurements of ground temperatures in South America at depths greater than 2 m, emphasizes the absence of an Andes-wide data repository akin to what is available for the northern hemisphere.

Given these limitations, a unique opportunity exists to advance knowledge of permafrost thermal state in South America through collaboration between researchers and private industry. This is especially true in the border area of Chile and Argentina, which holds significant reserves of precious metals and other natural resources at different stages of exploration, extraction, and development. Scientific investigations, often including the collection of ground temperature data in permafrost zones, are necessary to support environmental permitting and engineering designs. These investigations generate valuable data that can help assess the ground thermal regime in regions that have not yet been characterized and shared with the broader research community. In this study, subsurface thermal conditions along a North-South transect in the Central Andes (27°S-34°S) were examined by summarizing ground temperature data from a suite of boreholes installed by the private sector at eight distinct industrial project sites (Figure 1). The data were provided to the authors by BGC Engineering Inc., with permission of the individual project owners. The dataset was accompanied by confidential field notes and reports detailing instrumentation and site conditions to support the interpretations presented. All instruments were installed for environmental impact assessments or engineering design studies prior to the preparation of this paper.

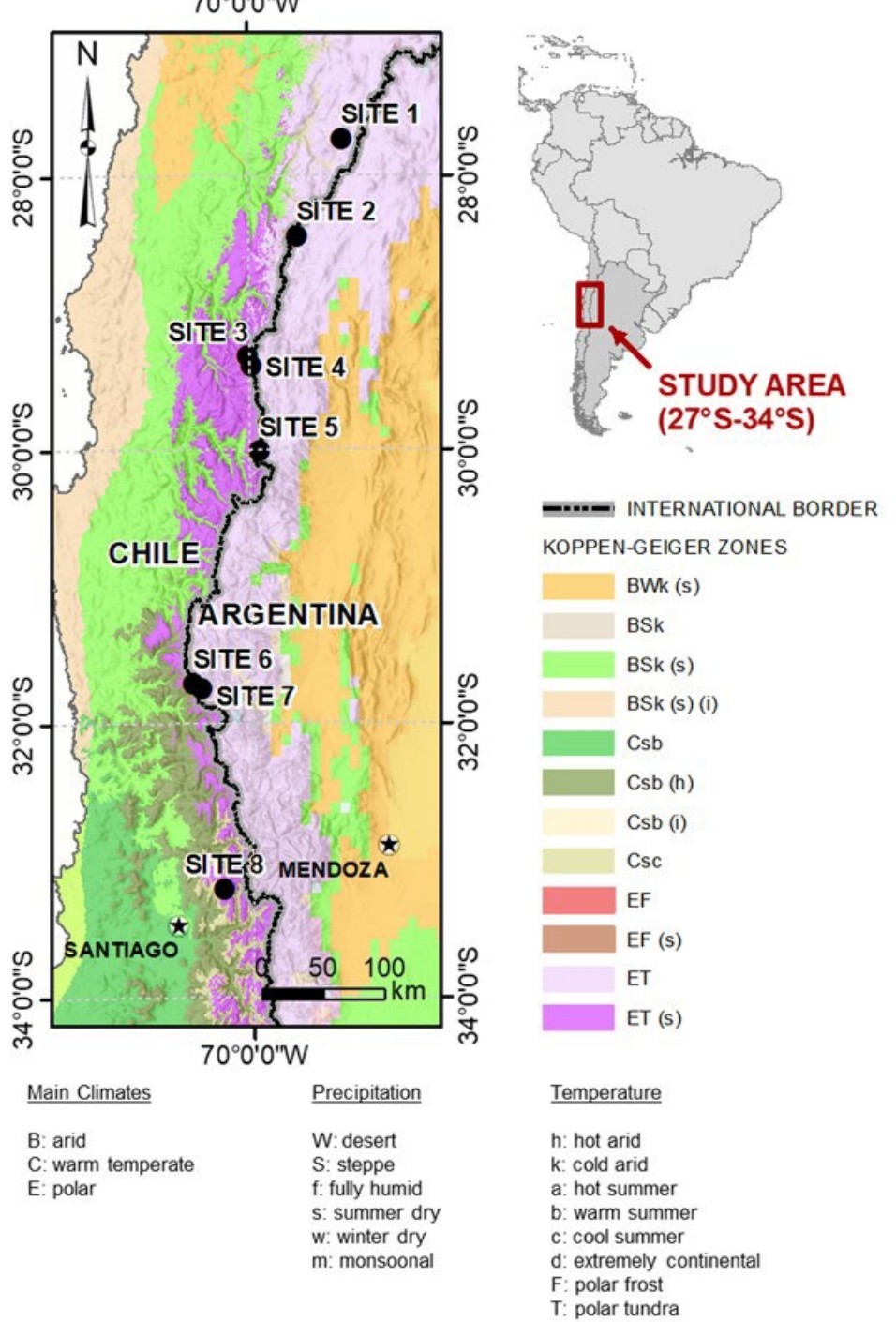

**Figure 1: Ground temperature monitoring locations and regional climatic setting. Koppen-Geiger zones are based on** Kottek et al., (2006) **for Argentina and** Sarricolea et al., (2017) **for Chile.**

| Citation | Site Location | Approx. Lat / Long | No. Boreholes | Landform / Surface Substrate | Monitoring Location Name | Max. Measurement Depth (m) | Elevation (m) | Monitoring Period | Comments |
|---|---|---|---|---|---|---|---|---|---|
| Trombotto et al. (1997); Trombotto & Borzotta (2009) | Cordón del Plata range (Argentina) | 32°54'S-33°01S / 69°27'W-69°15W | 2 | Morenas Coloradas Rock Glacier | Balcón I | 5 | 3,560 | 1989-1992 + 5 manual measurements collected in summers of 2004-2008 | Inactive thermokarst (average temperature ~0°C; seasonal variation correlates w with measured discharge; estimated deepening ~25 cm per year) |
| | | | | | Balcón II | 3 | 3,770 | 2001-2007 | Morenas Coloradas rock glacier, active thermokarst (permafrost deepening ~15 cm per year betw een 2002-2004) |
| DGA (2010) | Elqui River Catchment, La Laguna basin, Coquimbo (Chile) - Cerro Tapado, Topado Complex | 30°06'S-33°24S / 69°59'W | 2 | Llano de las Liebres Rock Glacier | Liebres 1 | 2.7 | 4,050 | Apr-Dec 2010 | Permafrost (if present) is below deepest sensor and estimated to be at least 3m deep |
| | | | | | Liebres 2 | 7.4 | 3,786 | Apr-Dec 2010 | Permafrost (if present) is below deepest sensor and estimated to be at least 8m deep |
| DGA (2010); Vivero et al. (2021) | Elqui River Catchment, La Laguna basin, Coquimbo (Chile) - Cerro Tapado, Topado Complex | 30°06'S-33°24S / 69°59'W | 2 | del Topado Rock Glacier | BH1 (Tapado 1) | 2 | 4,440 | 2011-2015 | Permafrost (if present) is below deepest sensor and estimated to be at least 3m deep |
| | | | | | BH2 (Tapado 2) | 2.4 | 4,405 | 2011-2015 | Permafrost (if present) is below deepest sensor and estimated to be at least 2.6m deep |
| Andres et al. (2011) | Chachani volcanic complex (Peru) | 16°11S / 71°31'W | 3 | Slope of Chachani Volcano | CHACHA-1 | 1.2 | 4,850 | 2007-2008 | -- |
| | | | | | CHACHA-2 | 1.2 | 4,976 | 2007-2008 | -- |
| | | | | | CHACHA-3 | 1.2 | 5,331 | 2007-2008 | Possible permafrost location |
| Monnier & Kinnard (2013) | Andes Rock Glacier, upper Choapa valley (Chile) | 31°48S / 70°30'W | 2 | Quebrads Noroeste Rock Glacier | DDH2010-1 | 25.0 | 3,767 | 2010-2011 | 5–7m thick active layer |
| Centro de Estudios Avanzados en Zonas Áridas | | | | | DDH2010-2 | 18.0 | ~3,800 | 2010-2011 | Measurements only performed 3 times per year |
| Atacama Ambiente (2017) | Goldfields Salares Norte Project (Chile) | 27°S / 68° W | 3 | -- | SNDD034 | 22.0 | 4,705 | Mar-Dec 2017 | Manual monthly measurements, non-permafrost borehole |
| | | | | | SNGET008 | 21.5 | 4,472 | Mar-Dec 2017 | Manual monthly measurements, non-permafrost borehole |
| | | | | | SNGET027 | 40.0 | 4,580 | Mar-Dec 2017 | Manual monthly measurements, permafrost from ~5-13 m deep |

**Table 1: Previously published ground temperature monitoring studies in permafrost regions of the Andes.**

**Table 1 (cont.): Previously published ground temperature monitoring studies in permafrost regions of the Andes.**

| Citation | Site Location | Approx. Lat / Long | No. Boreholes | Landform / Surface Substrate | Monitoring Location Name | Max. Measurement Depth (m) | Elevation (m) | Monitoring Period | Comments |
|---|---|---|---|---|---|---|---|---|---|
| DGA (2019) | Tupungatito Volcano (Chile) | 33°S / 69° W | 2 | Rocky ridge, near Tupungatito glacier | Tupungatito A | 2.0 | 5,575 | 23-Dec-19 | -- |
| | | | | Rocky ridge, part of a lava flow from the Tupungatito volcano | Sitio B, Tupungatito | 1.2 | 4,425 | 24-Dec-19 | -- |
| | Glaciar Bello (Chile) | 33°S / 70°W | 1 | 100 m NW of Bello Glacier | Bello | 1.7 | 4,840 | -- | No measurements published in report |
| Nagy et al. (2019) | Mt. Ojos del Salado (Chile) | 26°56'S-27°29'S / 68°32'W-69°15'W | 6 | Debris Covered Plateau | Laguna Negro Francisco | 0.6 | 4,200 | 2014-2016 | -- |
| | | | | | Murray Lodge | 0.35 | 4,550 | 2014-2016 | -- |
| | | | | | Atacama Camp | 0.6 | 5,260 | 2012-2016 | Possible permafrost borehole |
| | | | | | Tejos Camp | 0.6 | 5,830 | 2012-2016 | Possible permafrost borehole |
| | | | | | Crater Edge | 0.2 | 6,750 | 2012-2016 | Possible permafrost borehole |
| | | | | | Mt. Ojos del Salado summit | 0.1 | 6,893 | 2012-2016 | Possible permafrost borehole |
| Yoshikawa et al. (2020) | Coropuna volcanic complex (Peru) | 15°32S / 72°32'W | 15 | Hydrothermally Altered Lava | Coropuna-N3 | 0.3 | 5,694 | 2012-2019 | -- |
| | | | | | Coropuna-N2 | 0.3 | 5,564 | 2012-2019 | -- |
| | | | | | Coropuna-S4 | 0.3 | 5,100 | 2012-2019 | -- |
| | | | | | Coropuna-S3 | 0.3 | 5,053 | 2012-2019 | -- |
| | | | | | Coropuna-N1 | 0.3 | 4,886 | 2012-2019 | -- |
| | | | | | Coropuna-S2 | 0.3 | 4,711 | 2012-2019 | -- |
| | | | | | Coropuna-S1 | 0.3 | 4,247 | 2012-2019 | -- |
| | | | | | Coropuna-Borehole | 3 | 5,170 | 2012-2019 | -- |
| | Chachani volcanic complex (Peru) | 16°13S / 71°29'W | | | CHACHANI-CH3 | 0.3 | 4,711 | 2012-2019 | -- |
| | | | | | CHACHANI-Borehole | 4 | 5,331 | 2012-2019 | Possible permafrost borehole |
| | | | | | CHACHANI-Borehole surface | 0.02 | 5,331 | 2012-2019 | Possible permafrost borehole |
| | | | | | CHACHANI-South facing | 1 | 5,329 | 2012-2019 | Possible permafrost borehole |
| | | | | | CHACHANI-Higher albedo | 1 | 5,323 | 2012-2019 | Possible permafrost borehole |
| | | | | | CHACHANI-CH2 | 0.3 | 4,700 | 2012-2019 | -- |
| | | | | | CHACHANI-CH1 | 0.3 | 4,200 | 2012-2019 | -- |
| Mena et al. (2021) | Chajnantor Volcano, Atacama (Chile) | 22°59S / 67°43'W | 1 | Slope of Chajnantor Volcano | -- | | 14 | 5,640 | 2019 | Permafrost borehole, 5 m thick, gradient of 200°C/km; ALT ~ 14 cm |

## 2 Regional Setting

The study area is located in the Central Andes (27°S-34°S, Figure 1), where the climate is strongly influenced by the southeast Pacific anticyclone (SEPA), the Humboldt Current, and barrier effects of the mountains. Cold, humid westerlies associated with the Humboldt Current are diverted northward by the SEPA, while the Andes deflect Pacific air masses upwards and limit westward flow of moist air from the Amazon basin (Masiokas et al., 2020; Schulz et al., 2012). This creates the region's characteristic hyper-arid climate and orographic precipitation, which peaks in the austral winter (June to August) falling predominantly as snow. Average annual precipitation ranges between approximately 100 mm and 500 mm, with greater amounts generally observed at lower latitudes (Garreaud et al., 2020; Viale and Garreaud, 2015). The region experiences pronounced climatic fluctuations on interannual and interdecadal timescales due to interactions between the El Niño Southern Oscillation (ENSO) and the Pacific Decadal Oscillation (PDO) (Mantua and Hare, 2002; Montecinos and Aceituno, 2003; Schulz et al., 2012; Vuille et al., 2015; Garreaud et al., 2020). The Southern Annular Mode (SAM) also impacts climatic conditions on weekly to decadal timescales through modulation of south-westerly winds (Saavedra et al., 2018; González-Reyes et al., 2020; King et al., 2023). These climatic oscillations play a crucial role in shaping weather patterns and hydrological cycles in the Dry Andes, affecting agriculture, water resources, and ecosystems. The PDO influences long-term climate variability, with positive (negative) phases bringing warmer (cooler) and wetter (drier) conditions. ENSO strongly affects interannual climate variability, with El Niño events bringing warmer, wetter conditions, increasing snowfall in the Andes, while La Niña events lead to cooler, drier conditions. SAM variation influences interannual seasonal precipitation and snow cover in the Andes; positive phases are linked to reduced precipitation, and negative phases are associated with increased precipitation (Vera and Silvestri, 2009).

The eight project sites in this study are situated at high altitudes, ranging in elevation from 3,500 m to over 5,250 m above sea level. Each site exhibits significant topographic variability, with total relief ranging from 250 m to 1,200 m. The project sites fall within the 'Andean Arid Diagonal' (Bruniard, 1982), a contiguous zone of arid to semi-arid climate separating tropical and temperate climates of the northern and southern Andes (Figure 1). This includes parts of the Cold Mountain Desert (BWk) and High Mountain Tundra (ET) climatic belts (Kottek et al., 2006), which are distinguished by unique variations in altitude, precipitation, and temperature. Climatic conditions of the BWk belt, dominant below ~4,000 m are characterized by low humidity and minimal precipitation. With seasonal average temperatures ranging from ~18°C in January to ~8°C in July, permafrost will not form and is unlikely to exist in this zone; where it does exist, it is naturally degrading. Rock glaciers are the most common periglacial landform within the BWk belt and have variable ground ice content across the study area, reflecting different degrees of permafrost degradation (Hilbich et al., 2022). In contrast, the climate of the ET belt (elevations >4,000 m) is associated with low temperatures year-round and mean annual air temperatures (MAATs) below 0°C (Vuille et al., 2003; Garreaud, 2009), creating favourable conditions for permafrost formation. During the summer months (December to February), temperatures within the ET belt remain above 0°C, with daily highs exceeding 15°C and averages below 10°C.

Surface geomorphic indicators of permafrost in this belt, identified in the field and through remotely sensed imagery, include rock glaciers, gelifluction slopes, and patterned ground (Arenson and Jakob, 2010). In both climatic belts, snow accumulates at high altitudes during winter and remains until spring (October to November), followed by a unimodal snowmelt-dominated regime produced annually for all rivers originating from mountain peaks (Masiokas et al., 2016).

The distribution of permafrost across the Andes is complex, driven by high variability of mountain topography and climatic conditions at the catchment level. This leads to significant variations in ground thermal conditions over tens to hundreds of metres. Slight variations in altitude (usually spanning a few hundred metres) can generate marked differences in air temperature, precipitation, vegetation, snowpack, solar radiation, and glacial cover over short lateral distances (Arenson et al., 2022). Hyper-arid conditions, intense solar radiation and the southern hemisphere location creates nuanced variations in water and energy balances, shaping the distribution of permafrost in the Andes and distinguishing them from other mountain regions. The resultant effects on infiltration, subsurface freezing and thawing, and the movement of air and water through the ground contribute to the highly heterogeneous occurrence of permafrost in the region noted in several ground-based studies, with an even more complex distribution of ground ice (e.g., Hilbich et al., 2022). This is particularly evident in rock glaciers, where ground ice content varies widely from case to case (Arenson and Jakob, 2010; Hauck et al., 2011; Mollaret et al., 2020; Halla et al., 2021; Hilbich et al., 2022) and within individual landforms (Jones et al., 2019; Halla et al., 2021). In the southern Central Andes (32°S–36°S), the lower altitudinal limit of permafrost is estimated between 2,900 m and 3,200 m, with occasional occurrences up to 3,700 m (Saito et al., 2016). This varies with slope aspect due to variations in intensity of solar radiation, favouring the persistence of isolated patches of permafrost at lower elevations on pole-facing slopes compared to slopes oriented towards the equator (Yoshikawa et al., 2020; Arenson et al., 2022).

Several studies in the region have identified a progression from a cooler humid climate to warmer and drier conditions in recent decades (e.g., Carrasco et al., 2005; Falvey & Garreaud, 2009; Schulz et al., 2012; Jacques-Coper & Garreaud, 2015; Garreaud et al., 2020). A rise in air temperatures in the mid-1970s interrupted a slightly negative trend in maximum daily temperatures (Jacques-Coper and Garreaud, 2015). This rise was followed by a downward trend in maximum and minimum air temperatures into the early 2000s, coinciding with a cold-to-warm sea surface temperature (SST) shift of the PDO to its cool phase, possibly increasing El Niño events (Jacques-Coper and Garreaud, 2015; Schulz et al., 2012). Between 1979 and 2006, Falvey and Garreaud (2009) observed a cooling of 0.2°C/decade along the coast of Central and Northern Chile due to the Humboldt Current, while inland meteorological stations (Lagunitas and El Yeso) showed a warming of 0.25°C/decade. ENSO and IPO oscillations may have contributed to a decline in precipitation in coastal Chile in the late 20th century (Schulz et al., 2012), and the ongoing Central Chile Megadrought, which has been marked by a 25-45% precipitation deficit from 2010 to 2020 (Garreaud et al., 2020). Carrasco et al., (2005) documented a rise in the zero-degree mean annual air temperature (MAAT) isotherm in Central Chile from 1975-2001, with elevations rising by 122 m in winter and 200 m in summer. This shift facilitates snowmelt and a transition from solid to liquid precipitation, potentially triggering permafrost degradation (Zhang, 2005).

## 3 Methodology

### 3.1 Ground Temperature Data and Permafrost Presence

Between 2006 and 2017, ground temperature monitoring was systematically initiated at eight industrial project sites located between 27°S and 34°S and within approximately 25 km of the Chile-Argentine border (Figure 1). The compiled dataset presented in this study includes measurements collected from 53 boreholes distributed across the region, with 27 located in Chile and 26 in Argentina. The boreholes were installed to depths varying from 10 to 100 m at surface elevations ranging from 3,625 m to 5,251 m, in areas with and without permafrost. Of the total, 30 boreholes intercepted permafrost, while the remaining 23 were installed within unfrozen, or non-cryotic ground (Figure 2). Due to data interruptions and short monitoring duration, some boreholes in cryotic ground did not meet the two-year criterion of continuous ground temperature monitoring to confirm the presence of permafrost (van Everdingen, 1998). However, given that measurements were available from below the DZAA (i.e., from depths $\geq$ 10 m), it is reasonable to infer the presence of permafrost in boreholes where the temperature is at or below 0°C, even without a complete two-year record. Surface morphologies mapped at the borehole locations during thermistor installation include bedrock, colluvium, rock glaciers (containing ground ice), and landslide deposits (Table S1). Except for those in rock glaciers, the boreholes typically intercept bedrock within ~10-20 m of the ground surface.

Ground temperatures were monitored along the profile of each borehole using Negative Temperature Coefficient (NTC) thermistor strings (models YSI 4400-, RST TH00-, or Geoprecision TNode-series). Thermistors were accurate within a range of ±0.1 to ±0.5°C from base temperature, with precision ranging between 0.1°C and 0.25°C. Sensors were positioned at varying depths, starting from the ground surface (0 m) and spaced from 0.5 to 15 m (depending on the objectives and maximum depth of the borehole). Loggers were used for data collection at most locations although manual readings were made on occasion. The frequency of data collection varied with location, ranging from hourly to daily measurements.

The duration of monitoring varied by borehole, ranging from less than one month to nine years (Figure 2). Several locations experienced interruptions to data collection due to factors such as electrical storms, instrument malfunctions, inaccessibility for download or maintenance due to remoteness, adverse weather, slope instability and/or changing regulatory requirements. In some cases, interruptions occurred simply because a borehole was temporarily not being monitored as part of the project's objectives. A comprehensive discussion of data collection challenges, monitoring gaps, known data artifacts and filtering rationale is included in Section 3.2. Gaps in individual monitoring records were filled as outlined in Section 3.3 prior to interpretation of the data. Raw data for each borehole are presented in the supplementary information (Figures S1-S53).

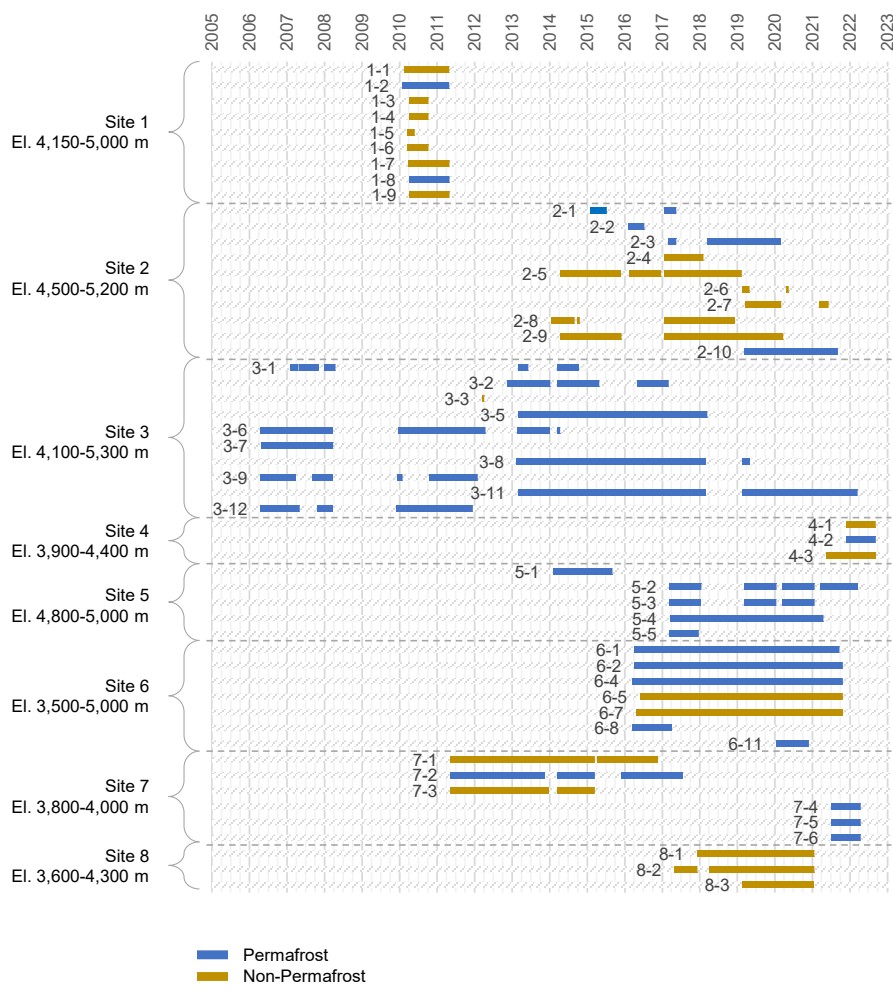

**Figure 2: Summary of available borehole temperature time series over time.**

## 3.2 Data Quality, Gaps and Filtering

Long-term operation of borehole infrastructure in high mountain regions can be hindered by natural factors including erosion, mass movements or meteorological effects, all of which may lead to instrument malfunction that can reduce data quality or create gaps in monitoring records. Challenging terrain, remoteness, and lack of financial support can limit site access, leading to poor instrument maintenance and data loss. Other factors affecting ground temperature monitoring unrelated to mountainous regions may include instrument damage by wildlife, vandalism, construction, or damage during instrument installation. Many of these challenges were encountered across the study area, with most interferences or malfunctions identified by field staff before analysis. The most common causes of data interruptions were battery power loss, faulty connections, or sensor failures between maintenance visits. Irregular funding for ground temperature monitoring based on specific project needs, led

to inconsistent and unpredictable field work and maintenance schedules, making it challenging to perform routine upkeep
(e.g., battery replacement of sensor repairs) proactively or in a timely manner.

Drilling activities were frequently identified as a source of interference to monitoring. While several thermistors were installed in pre-existing exploration boreholes, more than half of the locations were drilled explicitly for ground temperature monitoring. This included all boreholes at Sites 1, 4, 6 and 8, as well as boreholes 3-8, 3-11, 7-4, 7-5, 7-6 and 8-3. Dry sonic methods were used to drill all boreholes except at Site 1, which were advanced by diamond coring using water or polymer fluid. Early
temperature measurements in these boreholes may have been overestimated due to thermal disturbances from the drilling process, especially where fluid circulation was involved. Elevated temperatures at Site 1 are evident at the onset of monitoring, particularly from depths of 10 m and below (Figures S1-S9). Although temperatures gradually decrease as drilling disturbances dissipated, it remains uncertain whether ground temperatures at Site 1 had fully stabilized within the roughly one-year monitoring period. Early measurements from Site 1 and other locations with approximately one year of data (i.e., boreholes
6-8 and 6-11, 7-4, 7-5 and 7-6) were treated with caution in the analyses, particularly those collected during the initial two to three months, as they are unlikely to represent average ground temperature conditions. Drilling interferences were less of a concern for boreholes 3-8, 3-11, 8-3 and most boreholes at Site 6, which have longer monitored records.

At several monitoring locations, changes in ground conditions may have compromised measurement quality. For example, construction activities that altered local runoff patterns caused a large gully to form near borehole 5-1 (Figure 3). Field staff
first noted the gully in 2017, but anomalies in the data and multiple sensor failures suggested that erosional activity had affected borehole temperatures since late 2015. Consequently, measurements beyond September 2015 were excluded from analysis. Another instance of ground disturbance influencing monitoring was noted during a 2017 field visit, where a previously unseen debris flow was observed near borehole 2-2 (Figure 4). As with borehole 5-1, erroneous data were identified and excluded from analyses.

Project-driven optimization of monitoring sometimes led to data interruptions (or shortened record lengths) and informed additional filtering. For example, data collection ceased at select boreholes at Site 2 once permafrost was determined to be absent, and thermistor strings were relocated to higher elevations where encountering permafrost was more likely. Two thermistors were moved (from boreholes 2-4 and 2-5 to boreholes 2-6 and 2-7) resulting in the termination of monitoring at the original locations. During the relocation of the thermistor from borehole 2-4 to 2-6, it became evident that the sensor at
~24 m depth was damaged. Initially, anomalously high measurements recorded by this sensor at its original location were considered plausible, possibly due to warm groundwater or exothermic reactions at depth. However, similar anomalies persisted at the new location (borehole 2-6), suggesting measurements were artifacts and should be excluded from analysis. The supplementary information package accompanying this paper displays all raw data except for cases where artifacts were confidently identified and removed (e.g., related to erosion events or known instrument failures). Thermal disturbances from
drilling and occasional unexplained artifacts are visible in the figures but were omitted from analyses. These anomalies could be the result of water or air flow through blocky materials, or electrical storms which are common in central Chile

(e.g., Montana et al., 2021). As there was no clear indication that they were erroneous, these anomalies were retained on the plots for transparency and dataset completeness.

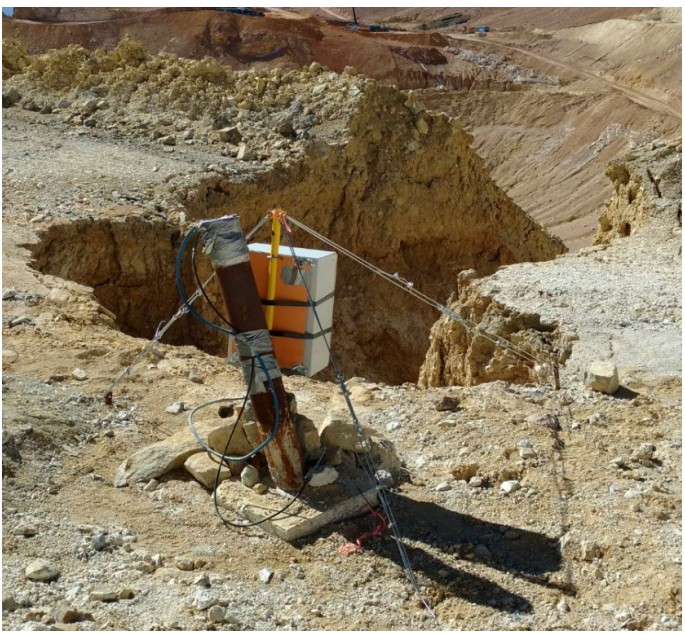

**Figure 3: Photograph of borehole 5-1 showing erosional gully that formed during the 2015 calendar year.**

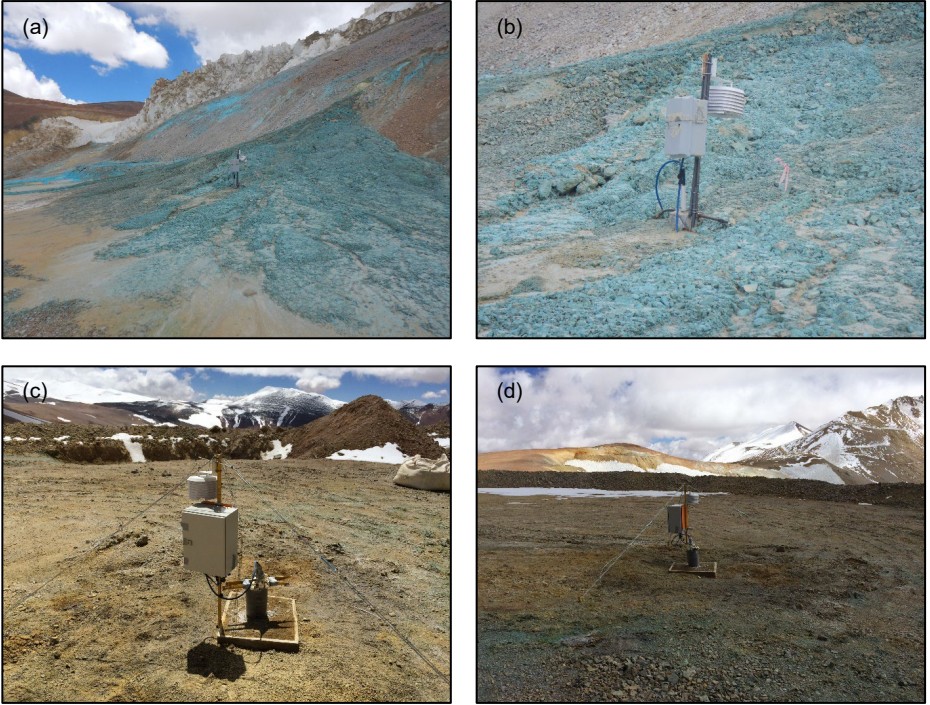

**Figure 4: (a) and (b) Original location of thermistor string at borehole 2-2 and the adjacent surface debris flows. The thermistor string was moved to boreholes 2-3 (illustrated in (c) and (d)) which is approximately 50 m higher in altitude.**

### 3.3 Filling of Data Gaps

Gaps in ground temperature time-series were interpolated using non-linear least squares regression to enhance data visualization and remove seasonal bias from analysis. At thermistors that showed seasonal variation, a sinusoidal function with a superimposed linear trend was fit to filtered data to approximate seasonal and longer-term temperature variations. For sensors located below the DZAA, linear interpolation was used. Initial conditions for each sensor were specified by setting function parameters (i.e., period, amplitude, phase shift, offset, slope) that produced a reasonable visual match to observed data.

Parameters were then optimized using the generalized reduced gradient (GRG) non-linear solver in Microsoft Excel to minimize the sum of squared residuals, targeting a normalized root mean square (NRMS) below 10%. The quality of fit was first assessed visually, with manual adjustments made occasionally to improve the overall match of the solution to the data. Missing values were then filled on a daily timestep using the fitted equation and are plotted alongside raw data in the supplementary information package (Figures S1 through S53).

It is noted that this approach has limitations in estimating temperatures within the active layer or in boreholes containing ground ice, as it cannot represent latent heat effects during phase changes. Additionally, it is not suitable for long records with complex warming or cooling trends, which may vary over the monitoring period. Despite these limitations, the method provided a useful way to estimate missing values and visualize seasonal patterns and short-term variations in the data, making it appropriate for this study, where data records at each instrument are less than 20 years.

### 265 4 Ground Thermal State

### 4.1 Seasonal and Interannual Ground Temperature Variations

A total of 22 permafrost and 12 non-permafrost boreholes have continuous records spanning at least one year, making them suitable for examining seasonal temperature variations (Figure 5). Most sensors near 10 m depth (Figure 5a) show clear fluctuations, with greater seasonal amplitudes in unfrozen ground (~0.5°C to 1°C) compared to frozen ground (<0.5°C). At

depths near 20 m (Figure 5b), sensors are mostly below the DZAA, with some exceptions in non-cryotic ground. Seasonal variations at both depth horizons are less pronounced where temperatures hover around 0°C, reflecting latent heat effects from annual freezing and thawing of ground ice. Similarly, attenuated temperature fluctuations at shallow depths (<2m) were noted during winter at 14 locations (Table S1, Figures S1 through S53) and are likely attributable to snow cover.

Unlike permafrost regions in the northern hemisphere, identification of any consistent trend of rising ground temperatures

within the Andean dataset to date is not possible due to its short duration. It instead reflects baseline local topo-climatic conditions and short-term climate fluctuations associated with the mountainous terrain. Both short-term warming and cooling are noted in the dataset, with no correlation to location, altitude or surface substrate (Figure 5). Figure 6 illustrates the wide

variation in short-term rates of temperature change from locations with at least two years of data (19 permafrost and 9 non-permafrost boreholes). Warming (0 to 0.05°C /yr) was noted in approximately half of the boreholes examined (15), with the rest showing short-term cooling. Non-permafrost boreholes exhibited greater variability in temperature change over the period of record.

Figure 6 includes representative long-term warming trends from other permafrost regions as documented in Smith et al. (2022). This includes continuous or "cold" Arctic permafrost (temperatures below -2°C, warming rates from ~0.04 to 0.11 °C/yr, monitored since the 1980s), discontinuous or "warm" Arctic permafrost (temperatures between -2° and 0°C, warming rates from~0.01 to 0.05°C /yr, monitored since the late 1970s to early 1980s) and mountain permafrost within the Swiss Alps (average: ~0.02 °C/yr, monitored since in the late 1980s to early 1990s). These trends are presented for reference only; but it is notable that the short-term warming rates in the Andes align with trends estimated for the Swiss Alps and the warm Arctic regions. However, when making this reference we strongly caution the reader, as northern hemisphere studies rely on significantly longer (decadal or multi-decadal) datasets than the limited records in this study. The longest record in the Andean analysis was approximately nine years (borehole 3-11), with most spanning between 2 and 7 years, making reliable comparisons with northern hemisphere studies premature. To establish causal relationships between ground temperatures and long-term climate variability in the Andes, ongoing monitoring of ground temperatures alongside local climate variables is needed. Given the constraints of the current monitoring dataset, a comprehensive analysis of this scale is beyond the scope of this study and presents an opportunity for climate researchers.

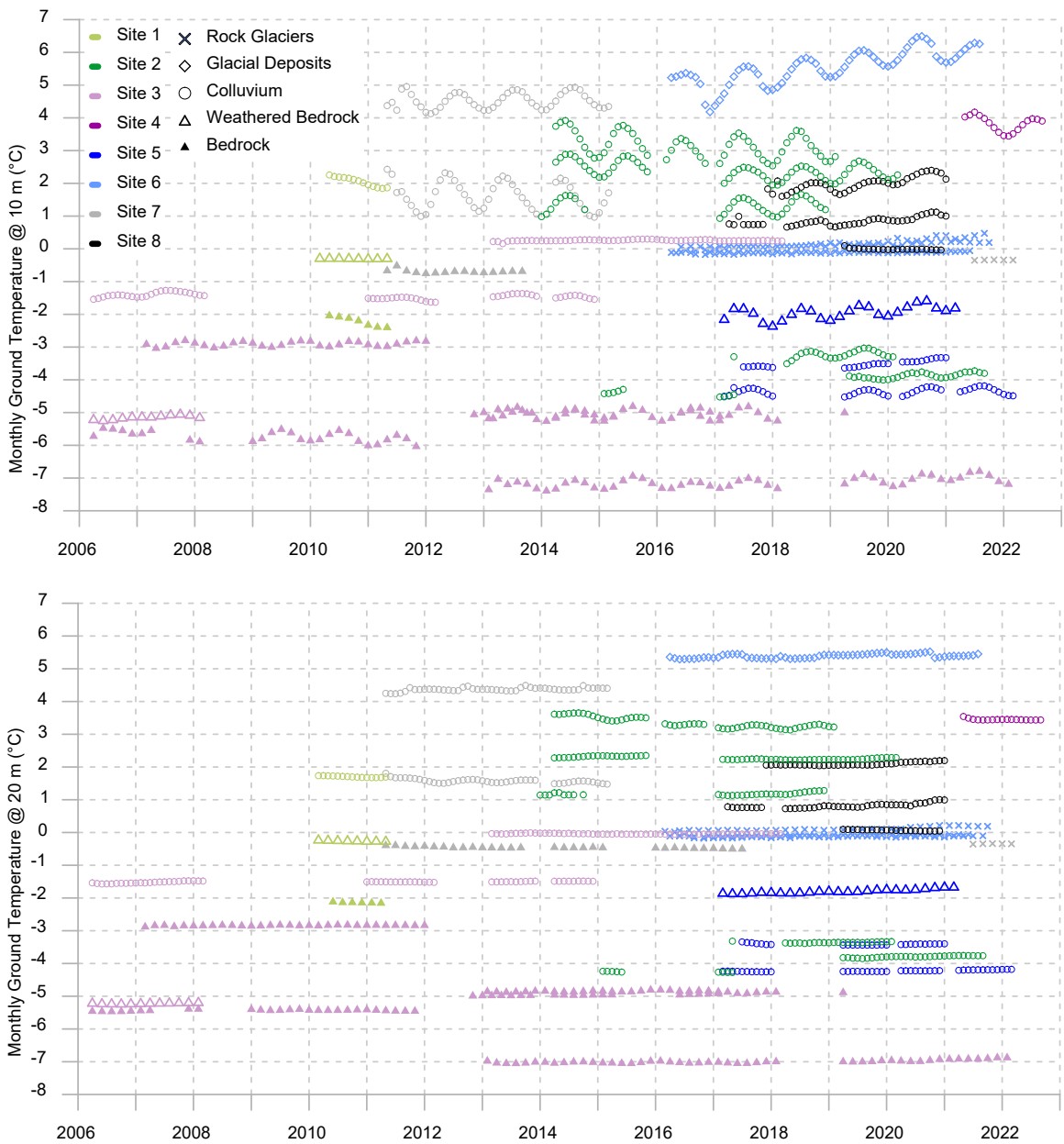


**Figure 5: Monthly ground temperatures at depths of 10 m (a) and 20 m (b) at boreholes with at least a full year of measurements. Measurements in bedrock and in rock glaciers are plotted every two months for clarity in the figure.**

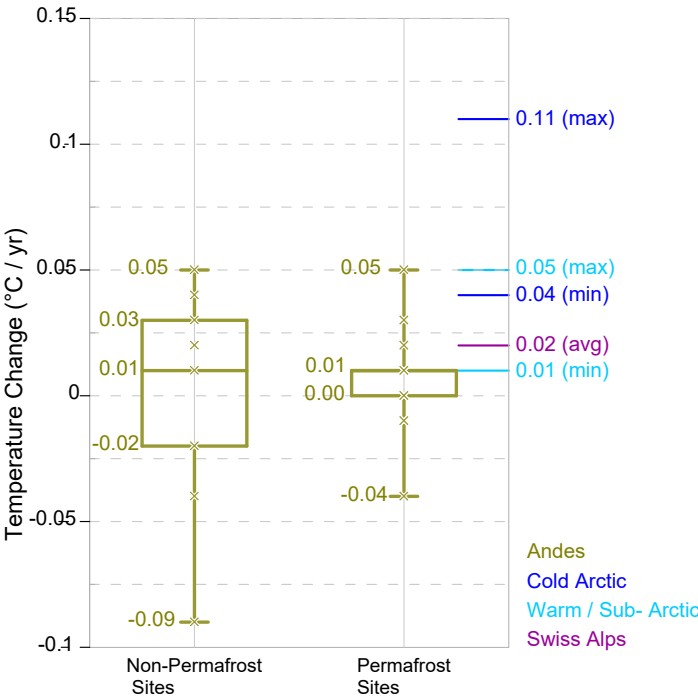

**Figure 6: Short-term ground temperature changes with time at 20 m depth in the Andes (this study), shown alongside long-term ground temperature trends compiled by Smith et al., (2022) in permafrost regions of the northern hemisphere (i.e., Cold Arctic, Warm / Sub-Arctic and Swiss Alps). The Andean estimates (n permafrost sites = 19; n non-permafrost sites = 9) are based on 2-9 years' worth of measurement and reflect short-term fluctuations in climate. Warming rates for other studies (n Cold Arctic Sites = 9; n Warm / Sub-Arctic Sites = 11; n Swiss Alps sites = 5) were derived from decadal / multi-decadal datasets and represent the effects of climate change. The terms "Cold Arctic permafrost" and "Warm/Sub-Arctic permafrost," from Smith et al. (2022), distinguish cold permafrost as below -2°C and warm permafrost as closer to 0°C. Estimates for the Andes sites are summarized in Table S1 with corresponding r$^2$ values.**

### 4.2 Depth to Permafrost

ALT in permafrost boreholes was estimated by linearly interpolating measurements between thermistors above and below the zero-degree isotherm at the time of maximum annual thaw. While this approach may slightly overestimate ALT (e.g., Riseborough, 2008), a lack of shallow temperature measurements at the Andes sites prevents reliable extrapolation of the zero-degree isotherm from above. Additionally, snow cover varies considerably across boreholes; in areas with little snow, atmospheric gradients strongly influence temperatures in the active layer, whereas snow-covered areas insulate the ground (e.g., BTS method by Haeberli (1978)). Given these complexities, and that the goal was to track potential changes in permafrost depth and compare boreholes, linear interpolation was considered a reasonable approach, allowing for relative comparisons rather than estimating absolute ALT values.

Of the 30 permafrost boreholes, 20 were considered in this analysis. Two rock glaciers (boreholes 6-1 and 6-4) were assessed as depth to permafrost table, as the depth to permafrost in these boreholes exceeded the maximum annual freeze/thaw depth. For boreholes 1-2 and 1-8, which lacked two consecutive years of data, the maximum annual thaw depth was estimated in a

similar manner to ALT (or permafrost table), as the records encompassed a complete freeze-thaw cycle with subsequent
refreezing. Permafrost locations with shorter monitoring records were not considered. Two permafrost locations (boreholes 3-7 and 3-12) were also excluded because the depth of thaw penetration remained above the shallowest sensors (< 1 m deep).

Figure 7 shows that thaw depth and ALT typically range from ~0.5 to less than 4 m, although ALT at boreholes 3-5 and 6-2 reached depths ≥ 6 m. Consistent with field observations of advanced degradation in these landforms (though not necessarily representative of the region), depth to permafrost within rock glaciers is the highest in the dataset, ranging from 7.3 to 17.4 m
in 2021. There is no consistent increase in ALT over time across the dataset, and in some locations, the active layer may be shallowing (e.g., boreholes 3-6 and 3-9, and possibly 2-10). In contrast, the top of permafrost is deepening in rock glaciers, at rates of approximately 0.4 m/yr at borehole 6-1, 0.8 m/yr at borehole 6-2 and 1.5 m/yr at borehole 6-4. These results reflect short-term fluctuations in climate.

Contour diagrams of borehole temperature evolution with time (Figure 8) reveal supra-permafrost taliks at boreholes 6-1 and
6-4. At borehole 6-4, the top of permafrost was fully decoupled from the active layer throughout monitoring. In contrast, the formation of the talik at borehole 6-1 began forming in mid-2019.

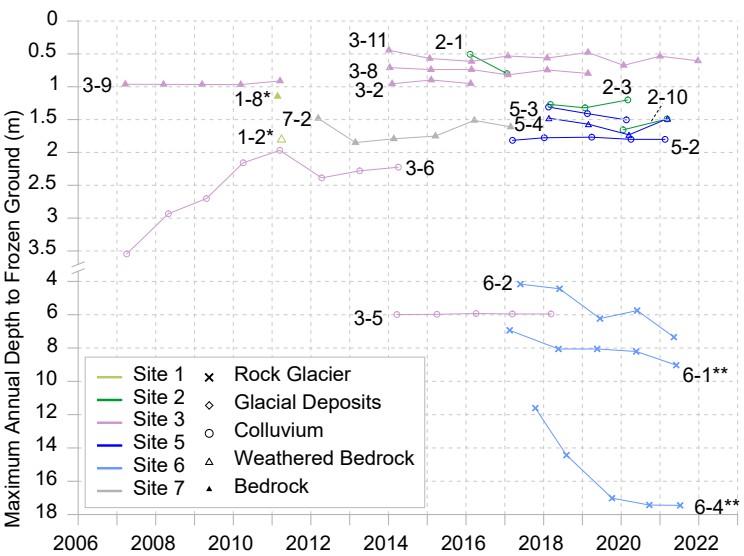

**Figure 7: Maximum annual depth to frozen ground within permafrost boreholes. Note scale break between 3.5 and 4m. \* indicates thaw depth (boreholes 1-2 and 1-8); \*\* indicates depth to permafrost table (boreholes 6-1 and 6-4). Active layer thickness (ALT) is**
**plotted for all other boreholes. See text for additional detail.**

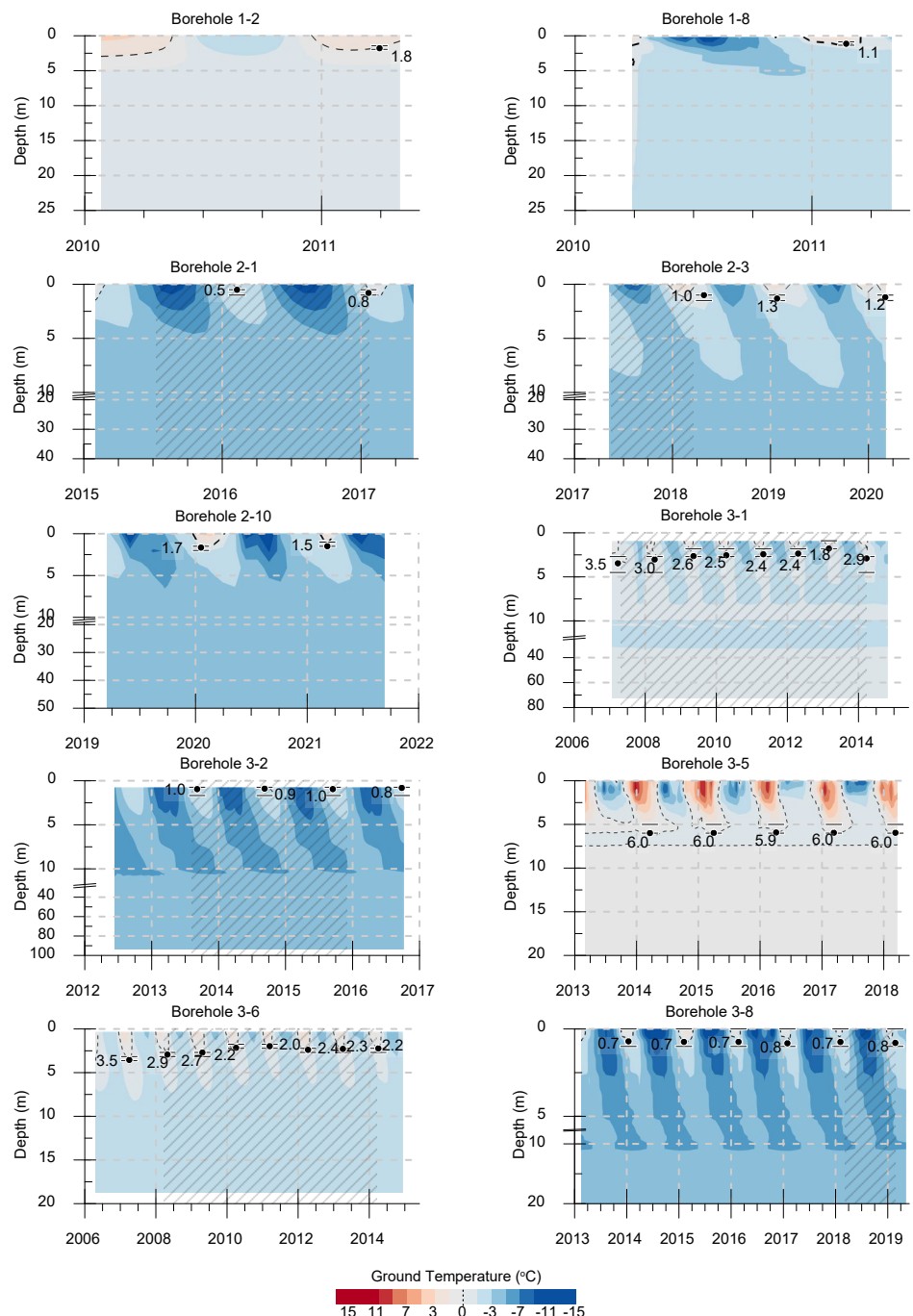

**Figure 8:** Interpreted borehole temperature evolution with time. Ground temperature contours estimated from raw data and gap-filled values; data gaps filled by interpolation are indicated by grey hatched areas. Depth to permafrost is indicated by black dots; horizontal lines indicate the depth of thermistors used to estimate permafrost depth. Note breaks in y-axes at selected boreholes for improve contour visualization.


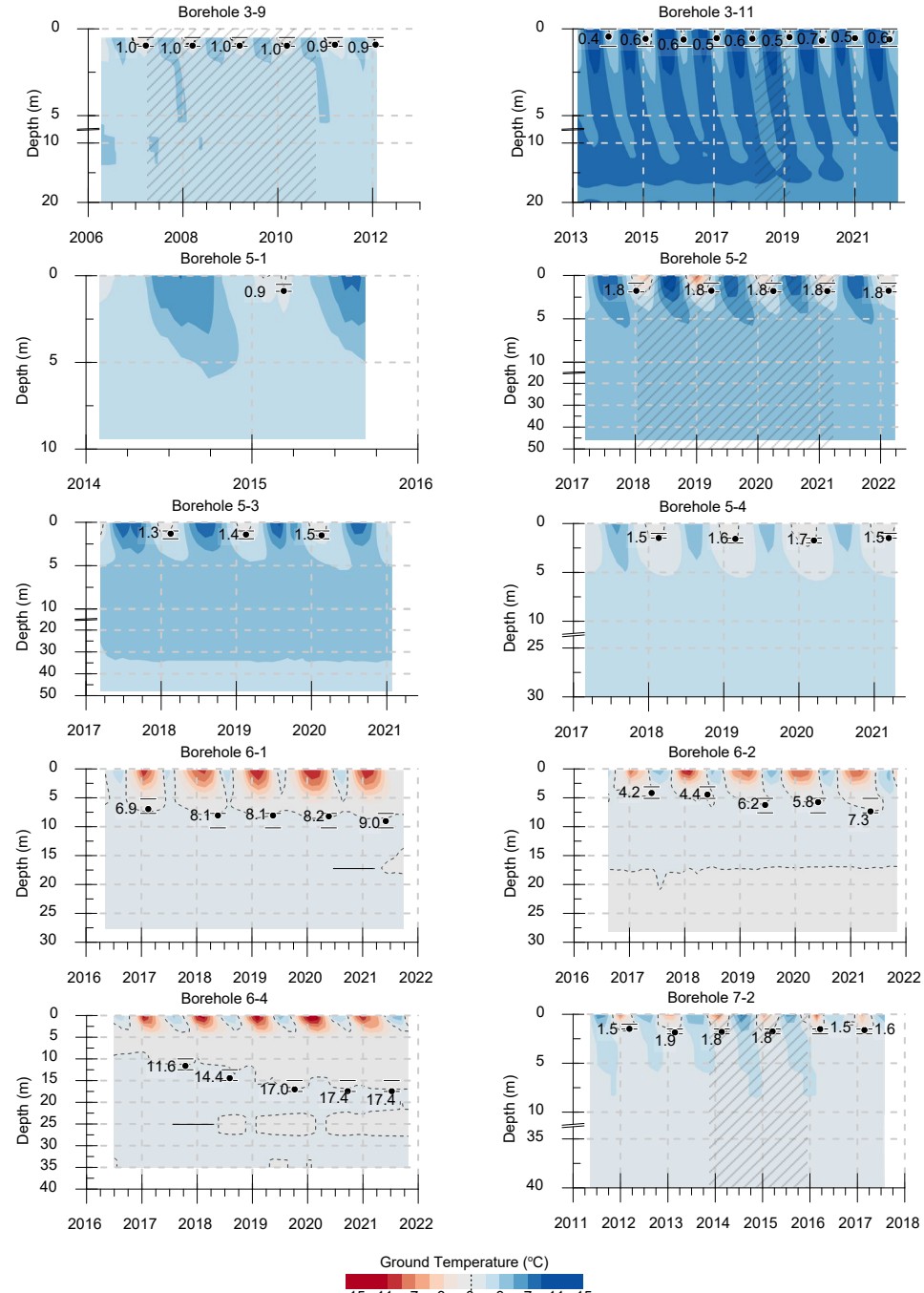

**Figure 8 (cont.): Interpreted borehole temperature evolution with time. Ground temperature contours estimated from raw data and gap-filled values; data gaps filled by interpolation are indicated by grey hatched areas. Depth to permafrost is indicated by black dots; horizontal lines indicate the depth of thermistors used to estimate permafrost depth. Note breaks in y-axes at selected boreholes for improve contour visualization.**

## 4.3 Ground Temperature Profiles

The temperature profile within a borehole is shaped by factors such as regional geothermal heat flux, lithology variations, local topo-climatic variations, and historical fluctuations in ground surface temperatures. These factors result in a wide range in
ground temperatures (from approximately -7 to 7°C) and varied profile curvatures (Figure 9), reflecting the complex thermal landscape of the Andes. Variations in profile curvature within individual boreholes suggest ground temperatures are not in equilibrium with modern climate conditions. Instead, they represent the present balance between surface and geothermal heat fluxes controlled by thermal properties of the ground, which vary with location and depth. Borehole temperature gradients within the upper 30 m are both positive (warming with depth) and negative (cooling with depth), with some locations showing
nearly isothermal conditions. Several boreholes in warm permafrost (3-5, 6-2, 6-5, 6-8, 6-11, 8-3) exhibit a composite temperature profile within the upper 30 m, characterized by a nearly isothermal region close to 0°C, below which temperatures increase with depth due to the geothermal heat flux. The temperature profiles shown in Figures S1 through S53 reveal the presence of thin permafrost layers in several rock glaciers, ranging from approximately 2 to 40 m thick. Permafrost thickness in boreholes that did not intercept the base of permafrost (but exhibited warming with depth) was estimated to range from 40 to
> 500 m, based on the projection of thermal gradients to the zero-degree depth intercept.

There is no clear relationship between profile shape or permafrost thickness and latitude (i.e., site number), likely due to the wide variation in ground elevations and surface slope orientations within a few tens of metres at each project site (Table S1 and Section 3.3). Such topographic heterogeneity results in significant variations in surface solar radiation and microclimatic conditions, which have a greater influence on ground temperatures than latitude. There also does not appear to be a relationship
between profile curvature or permafrost thickness with surface morphology, likely because most boreholes intercept bedrock at depths of 10-20 m (BGC Engineering Inc, Pers. Comm. October 2023). As such, temperature profiles generally reflect thermal properties of shallow bedrock, with minimal influence from surface substrate.

Potential exceptions to this are noted in rock glaciers at Site 6 (i.e., boreholes 6-2, 6-5, 6-8 and 6-11), which, as noted above, are characterized by shallow isothermal conditions near ~ 0°C then increasing temperatures with depth. The coexistence of air
and ice in the pore space near the ground surface and within the active layer of these coarse blocky landforms results in a significantly lower thermal conductivity compared to bedrock. In addition, the relatively large pore space promotes air convection, which can significantly cool the active layer, leading to temperatures ~1°C colder than without convection (e.g. Wicky and Hauck, 2020). Below the active layer, the ice-rock mixture is likely to be less sensitive to variations in the near-surface thermal regime (although still distinct from bedrock), leading to a unique profile shape despite possibly similar thermal
histories at the ground surface. As these particular landforms are known to contain ice-rich permafrost near the phase change temperature, vertical segments of the profiles may reflect the melting of ground ice, slowed due to latent heat effects.

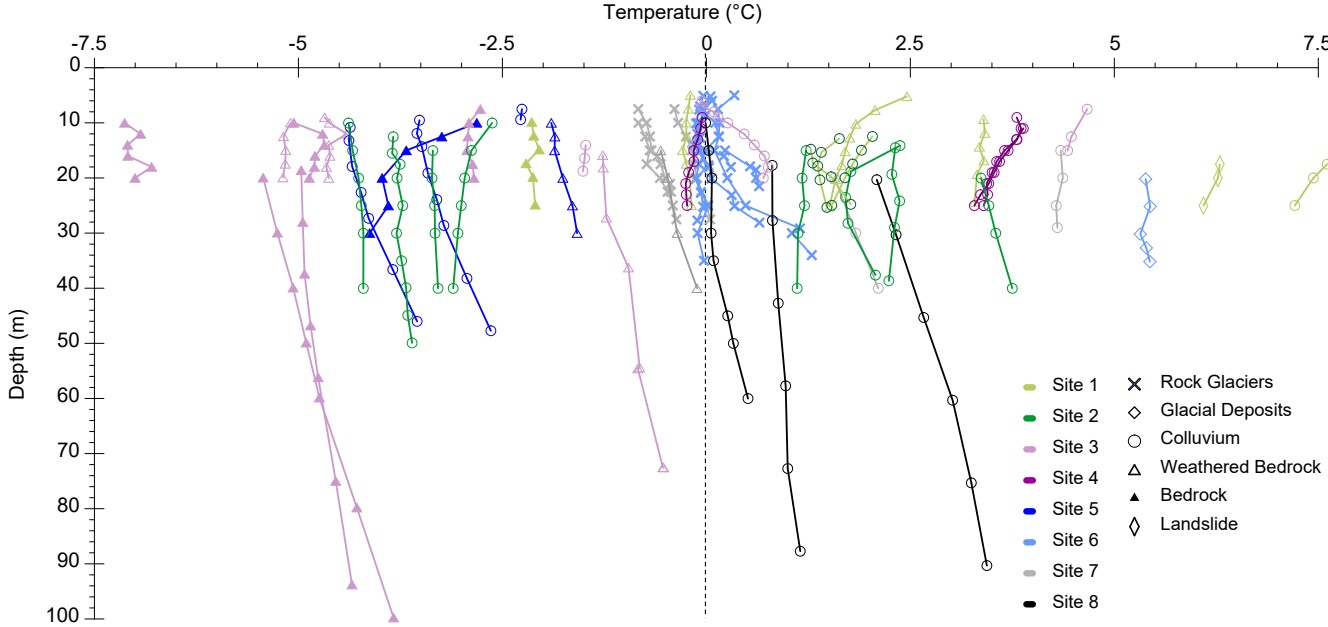

**Figure 9: Ground temperature profiles at all boreholes. Average and most recent values shown for locations with ≥1 year and <1 year of data, respectively. Excludes shallow measurements (between ~5-20 m) influenced by seasonal temperature variations.**

### 4.4 Thermal Gradients

Thermal gradients of individual boreholes provide insight into the recent thermal evolution of the ground in response to changing surface temperatures. The attenuation of historical surface temperature anomalies may be indicated by warm (cool) deviations of shallow borehole temperatures from the linear trajectory of the deep gradient. This temperature deviation (or offset) was estimated as the difference between projected surface temperatures from shallow (< 30 m) and deep (30-100 m) gradients. Offsets estimated in this depth range reflect recent decade-scale shifts in surface temperatures, assuming equilibrium conditions and uniform thermal properties with depth (e.g., Lachenbruch and Marshall, 1986). Since thermal responses to surface warming at depths greater than ~100 m likely lag several decades, the offset provides a first approximation of secular temperature changes with time.

Nineteen boreholes (10 permafrost and 9 non-permafrost boreholes) in the Andean dataset extend to depths beyond 30 m and were considered in this analysis. Rock glaciers were excluded due to their unique thermal profiles (Section 4.3), as were boreholes 2-4 and 2-6 due to erroneous measurements at ~24 m (Section 3.2). Thermal gradients were estimated from the linear segments of profiles at intermediate depths (below the DZAA to 30 m) and from 30 m to the final depth of each borehole, except at borehole 8-1. The shallower gradient at borehole 8-1 was estimated from the DZAA to 60 m, and the deep gradient from 60 m onward due to a slight decline observed in the thermal gradient at approximately this depth (Figure 9).

Results of the gradient analysis (Figure 10) are presented in a similar manner as the temperature changes derived from the interpolation of time-series data (Figure 6) for conceptual comparison of the two analyses. As with the time-series analysis,

the gradient analysis indicates both warming and cooling in recent history, with greater variability in the non-permafrost boreholes. Warm-side deviations from deep thermal gradients are evident for 14 of the boreholes (7 permafrost and 7 non-permafrost), and range from 0.05 to 0.9°C. The remaining boreholes (2 permafrost and 3 non-permafrost) indicate cooling in recent decades, with offsets ranging from approximately -0.05 to -1.08°C. It is emphasized that these estimates are derived from many simplifying assumptions, and that three-dimensional analysis accounting for ground thermal properties are necessary for a more accurate understanding of near-surface ground temperature changes in recent history.

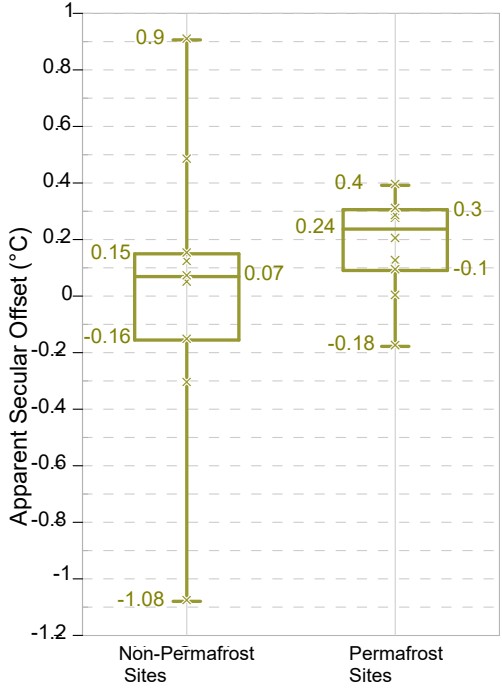

Figure 10: Variation in apparent secular offset in ground surface temperatures, estimated from gradient analysis (n permafrost sites = 19; n non-permafrost sites = 9).

### 4.5 Aggregate Trends

Representative ground temperatures across the study region show no correlation with latitude, but significant variation occurs over short lateral distances at the site level (Figure 11). Transitions from cold to warm permafrost (< -2°C to 0°C) and/or to non-cryotic ground occurs with 1 km, highlighting the greater influence of catchment-scale topo-climatic variability over regional climate. Ground temperatures generally decrease with increasing altitude, showing no clear relationship with surface morphology, as most measurements represent thermal conditions of shallow bedrock (Figure 12a). Rock glaciers, however, consistently show lower temperatures than boreholes at similar elevations (≤ 0 °C in rock glaciers vs. 2-6°C in other boreholes). They also mark the lowest altitudinal occurrence of permafrost within the dataset (slightly below 3,600 m) due to a sustained presence of ground ice compared to the broader dataset. Excluding rock glaciers, ground temperatures correlates more strongly

with altitude at permafrost boreholes ($r^2 = 0.65$) than for the complete dataset ($r^2 = 0.53$), with lapse rates of approximately - 4.3°C/km and -5.7°C/km, respectively.

Consistent with cooler temperatures at higher altitudes, both depth to permafrost and permafrost thickness decrease as elevation increases (Figure 12b and c). Again, no correlation with surface morphology is evident except in rock glaciers, which show the most variable depth to permafrost and lowest estimated thickness within the dataset, reflecting the advanced state of degradation of these landforms (e.g., supra-permafrost talik at boreholes 6-1 and 6-4, Figure 8). Excluding rock glaciers, the depth to top of permafrost decreases by approximately 1.9 m/km elevation gain ($r^2 = 0.30$), while permafrost thickness increases at approximately 300 m/km ($r^2 = 0.45$).

Incorporating slope aspect into the analysis reveals that the zero-degree isotherm for ground temperature occurs at higher elevations on northeast-facing slopes, ranging from below 3,700 m (within rock glaciers) to approximately 5,000 m (Figure 13a). This asymmetry is partly due to variations in average incident solar radiation with aspect (Figure 13b), and more broadly to the geographical position of the study area within the southern hemisphere. Coincidentally, all monitoring locations within rock glaciers are on south-facing slopes, where lower solar radiation may help sustain permafrost in these landforms.

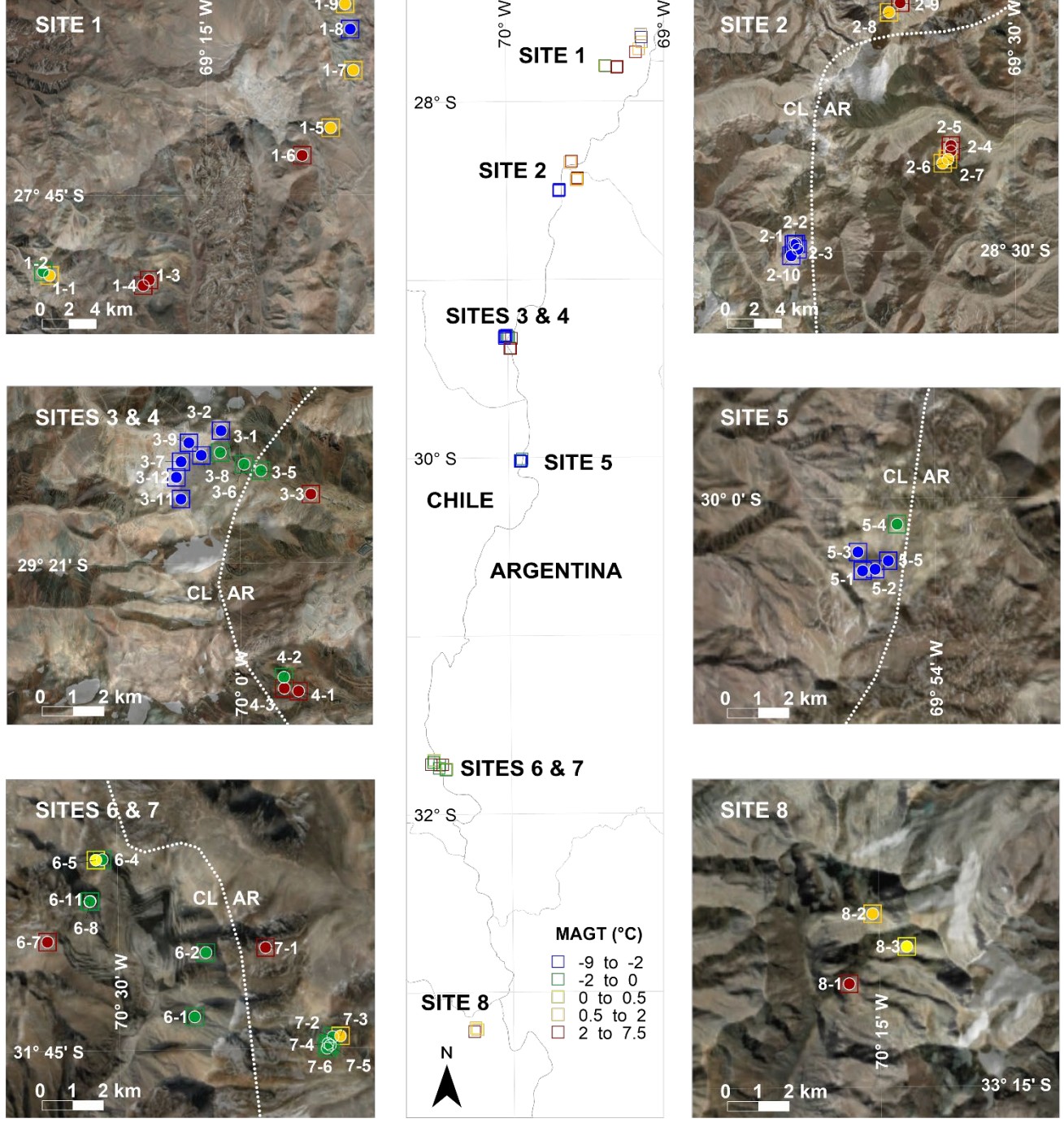

Figure 11: Representative ground temperatures across the study area. Measurements are shown for depths of 20 m and/or within permafrost zones of the boreholes (boreholes 3-5, 6-2, 6-4, 6-8 and 6-11), or from the deepest sensor if the thermistor string was shorter than 20 m (boreholes 3-3 and 5-1). Temperature values and measurement depths are summarized in Table S1.

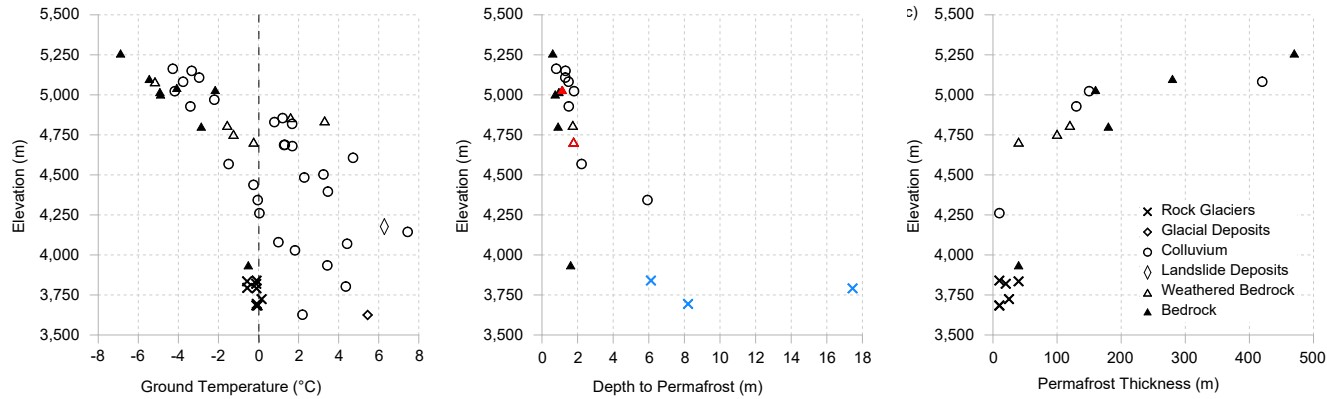

**Figure 12: Relationship of (a) ground temperatures and (b) depth to permafrost and (c) permafrost thickness with altitude. Thaw depth and depth to permafrost table coloured red and blue on diagram (b). The remaining points on diagram (b) represent the thickness of the active layer (ALT).**

**Figure 13: Aspect-elevation diagrams showing variations in (a) ground temperature and (b) average incident solar radiation. Aspect was estimated from ASTER GDEM v3 Worldwide Elevation Data (1 arc-second Resolution). Solar radiation estimated using ESRI Area Solar Radiation toolset. (https://desktop.arcgis.com/en/arcmap/latest/tools/spatial-analyst-toolbox/area-solar-radiation.htm).**

# 5 Discussion

## 5.1 Significance of the Dataset

This study provides new insights into permafrost dynamics in the Central Andes, some which were previously hypothesized based on studies from other permafrost regions but lacked sufficient data to confirm their broader relevance to South America.

It represents the largest and most regionally extensive compilation of ground temperature data from high altitude (>3,500 m) sites with permafrost in South America to date, filling a critical knowledge gap in permafrost research. Using data from 53 boreholes (10 m to 100 m deep) within permafrost and non-permafrost zones, the compilation provides a unique snapshot of ground thermal state within the Andean Cordillera of Chile and Argentina. Adequate depth of monitoring throughout the dataset enables characterization of average temperatures below the DZAA, visualization of borehole thermal gradients, and

first-order estimates of the permafrost thickness and depth. This level of insight surpasses that of the few key existing monitoring studies in the Andes, which generally are limited to boreholes just a few metres deep (Trombotto et al., 1997; Trombotto and Borzotta, 2009; DGA, 2010; Andrés et al., 2011; DGA, 2019; Nagy et al., 2019; Yoshikawa et al., 2020; Mena et al., 2021; Vivero et al., 2021).

An important implication of this work is that the data can be used to validate existing permafrost distribution models in the

455 region, which were previously developed without any borehole temperature data (e.g., Arenson and Jakob, 2010; Ruiz and Trombotto, 2012). Additionally, having ground temperature data within permafrost and non-permafrost zones helps reduce the risk of site-selection bias towards permafrost presence, which can complicate evaluations of spatially distributed models predicting permafrost. Beyond validating existing permafrost distribution models in the region, data from this study may be utilized to support upscaling endeavors like those of Mathys et al. (2022), to quantify ice content in Andean permafrost regions

and inform future water resource planning amidst the challenges posed by climate change.

Several important insights can be derived from the presented dataset that broaden the understanding of permafrost thermal state in both the Andes and the global permafrost context. Some of these findings are consistent with observations in other permafrost regions, while others may uniquely represent the Central Andes, or more precisely, the region within the altitudinal and latitudinal constraints of the dataset (i.e., 3,625 m to 5,251 m and 27°S-34°S). These are discussed further in the sections

that follow.

## 5.2 Temporal Variations in Ground Temperature

The Andean data compilation shows no consistent trend of warming in recent history. Instead, warming and cooling of the ground are inferred from time series (Section 4.1) and thermal gradient analysis (Section 4.4). Limited by the short monitoring period (<10 years), the time series analysis captures short-term fluctuations driven by seasonal and inter-annual climate

variability and the study area's mountainous topography. While thermal gradient analysis may offer more reliable decadal-scale estimates of ground temperature change, its failure to show a consistent trend could stem from oversimplified assumptions about borehole geology and varying climatic conditions across locations.

Due the short record length, geologic complexity and topo-climatic variability between sites, it is not possible to determine whether the ground thermal regime in the study area follows long-term warming trends observed in other permafrost regions. However, it is worth noting that similar deviations from long-term trends have been observed in global datasets, and are linked to short-term local meteorological influences (e.g., Biskaborn et al., 2019; Etzelmüller et al., 2020; Haberkorn et al., 2021). Localized changes in snow cover, vegetation, and soil moisture content, especially in mountain environments, are known to significantly impact the thermal state of the ground, as evidenced by periods of permafrost cooling in the Alps in 2016 and 2023, attributed to anomalously low snow conditions during those years (PERMOS, 2023). Also in the Alps, summer heat waves during 2016 and 2019 were shown to correlate with short-term increases in active layer thickness (PERMOS, 2023), and water percolation has been linked to accelerated permafrost degradation (Luethi et al., 2017). Interannual variability in snow cover is inferred from near-surface isothermal conditions within the active layer at several of our Andean boreholes (Figures S1 through S53). The resulting irregular insulation of the ground during winters, along with inconsistent infiltration of meltwater in the spring, likely contributes to both temporal and spatial variability of ground temperatures in the dataset, although latent heat released from the annual freezing and thawing of porewater may also play a role. The significant dryness of the Central Andes compared to other mountain environments, which has been exacerbated in recent years by the megadrought (Garreaud et al., 2020), also leads to comparatively less water available from snowmelt to infiltrate the ground and promote permafrost degradation. Long-term warming trends may also be obscured by temporary, localized surface temperature inversions, which can cause air temperatures at lower elevations to be cooler than expected, and vice versa. Such inversions have been shown to create unique permafrost conditions in near-proximity dissimilar valleys in Yukon, Canada, where ground temperatures at some high-altitude sites are significantly warmer than those at adjacent valley bottoms (Lewkowicz et al., 2012; Noad and Bonnaventure, 2023).

At a more regional scale, the combined influence of SAM, ENSO and PDO on South American climate patterns (e.g., Mantua and Hare, 2002; Montecinos and Aceituno, 2003; Vuille et al., 2015; Saavedra et al., 2018; Garreaud et al., 2020; Yoshikawa et al., 2020; González-Reyes et al., 2020; King et al., 2023) can temporarily influence the ground thermal regime across the study area. This would occur primarily through changes to spatial and temporal snowpack distribution, snow-albedo feedback, and variations in water infiltration that impact latent heat absorption. Although temperature and precipitation patterns directly related to SAM, ENSO and PDO and their potential impacts to the ground thermal regime were not examined in this paper, some patterns are apparent for the Central Andes and may be applicable to the study area. This includes temporary cooling trends observed in shallow (2 m deep) ground temperatures simultaneously with deceleration of several rock glaciers at a site with permafrost in Central Chile (within the range of this study) aligning with lower MAATs during the same period (2010-2015). These cooling trends were linked to the predominance of La Niña and neutral ENSO conditions since 2009 (Vivero et al., 2021).

With a monitoring record that currently falls short of the ideal length to assess impacts of atmospheric warming on ground temperatures (i.e., 20 years or more), air temperature data collected in the study region may offer complimentary insights to the observations presented in this work. A summary of MAATs in mountainous regions in South America by (Hock et al.,

2019), which is based on very limited monitoring data, indicates lower warming rates or even slight cooling trends when compared to global studies. One meteorological station located at Site 3 (~5,000 m, ~29°S) demonstrates relatively stable air temperatures over 20 years of monitoring (1999-2021, Figure 14). This apparent stability and/or slight lowering of MAATs in South America suggests that the trajectory of ground temperatures in the Andes is likely to be unique from other regions that show clear signs of warming. The unique topo-climatic attributes of the Andean cryosphere – characterized by arid conditions, high solar radiation, minimal vegetative cover and organic matter, and less massive ice (except for rock glaciers) together with mountain topography – may expedite energy transfer and reduce latency of temperature change compared to other permafrost regions, both in mountains and the Arctic. Regardless of the specific mechanisms driving ground temperature evolution in the Andes, the preceding discussions emphasize the importance of continued monitoring to establish robust connections with climate change. This effort must also consider local climate and geographic conditions, as well as relationships with oceanic phenomena.

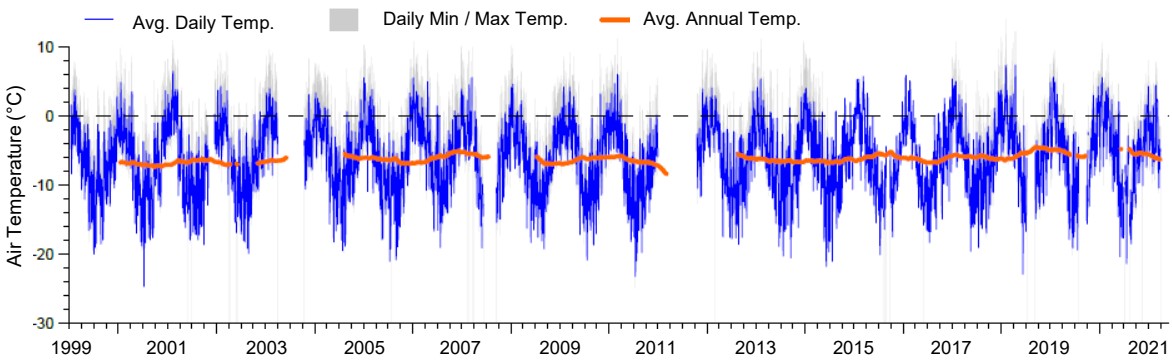

**Figure 14: Air temperature time series collected at Site 3 (El 4,927 m). Shows stability of temperature data over 20 years. Source: BGC Engineering Inc. (2021).**

**5.3 Shared Characteristics with other Mountain Permafrost Regions**

Although the topo-climatic conditions of the Andes are unique among permafrost zones, some common characteristics with other mountain environments can be discerned from the data compiled in this study. High spatial heterogeneity in ground temperatures, correlations with altitude and slope aspect, and distinct thermal characteristics of rock glaciers compared to other landforms, suggest that similar processes are shaping the current and changing ground thermal regime in the Central Andes. Significant variations in ground temperature over short horizontal distances - shifting from cold to warm permafrost (< -2°C to 0°C) and/or non-permafrost ground within less than 5 km - emphasizes the strong microclimatic influence on the distribution of permafrost at catchment level over regional climate stressors. The pattern of decreasing ground temperatures, greater depth to permafrost, and increasing permafrost thicknesses with increasing altitude, reflects the orographic influence of air temperatures, a common feature of mountainous regions.

The spatial pattern of ground temperatures with respect to slope aspect reflects the asymmetry of incident solar radiation of the geographic region, resulting in a higher elevation of the zero-degree ground temperature isotherm on northeast facing slopes. These characteristics were previously inferred for the Central Andes based on regional climate and mapping studies (e.g., Saito et al., 2016) and general knowledge of mountain permafrost environments (e.g., Haeberli, 1973). However, until now, these assumptions were not extensively validated with ground temperature data specific to the Andes.

Rock glaciers emerge as thermally unique permafrost landforms in the Andean dataset, a characteristic that has been widely observed in other mountain permafrost regions (e.g. Barsch, 1977). This uniqueness arises from their coarse blocky morphology and high porosity, which enable substantial variations in air, water, and ground ice content - factors known significantly influence ground temperatures, both spatially and temporally. This effect is particularly pronounced when rock glaciers contain substantial ground ice near the phase change temperature (e.g., Haeberli et al., 2006). Ground temperatures within rock glaciers in the Andean dataset do not follow general altitudinal relationships of other boreholes—instead, they consistently show lower temperatures than other boreholes at similar elevations, likely due to reduced thermal conductivity, convective cooling in the active layer and the sustained presence of ground ice from latent heat. Depth to permafrost and permafrost thickness in rock glaciers also deviate from the broader altitudinal relationships observed in the dataset, exhibiting the greatest and most variable depths to permafrost, and lowest permafrost thickness within the dataset. Within rock glaciers, permafrost depth was estimated to be > 7 m and permafrost thicknesses ranged between 2 m to 40 m (compared to depths < 4 m and thicknesses typically > 40 m in other boreholes). The formation of a supra-permafrost talik at two locations (boreholes 6-1 and 6-4) reflects an advanced state of degradation in these landforms.

Finally, rock glaciers appear to mark the lowest altitudinal limit of permafrost in the Andean dataset, with the lowest measurements slightly below 3,700 m at Site 6. The remaining measurements within rock glaciers cluster near 0°C between elevations of 3,700 m and 3,800 m, and other permafrost boreholes are located at altitudes > 4,250 m. Although this estimate roughly aligns with previously reported limits in the region (i.e., ranging between 2,900 m and 3,200 m, and occasionally reaching elevations of 3,700 m (Saito et al., 2016)), caution is warranted due to sample coverage bias associated with industry-driven data collection, rather than permafrost delineation objectives. The same caution applies to the asymmetry of the interpreted zero-degree ground temperature isotherm with respect to aspect for the full dataset.; while rock glaciers occur on south-facing slopes and in areas that experience lower solar radiation, their absence on north facing slopes may partially be a consequence of sampling bias rather than solar radiation effects.

**6 Conclusion**

This study presents the first regional compilation of in-situ ground temperature data from mountain permafrost regions of the Central Andes (27°S-34°S) at depths below seasonal influences. Compiled from 53 boreholes along a 650 km-long north-south transect near the Chilean-Argentine border, the dataset offers new insights into regional baseline thermal conditions and permafrost temperature dynamics in South America. The analyses highlight similarities with ground temperature

characteristics of other mountain permafrost regions, while also revealing unique aspects of the ground thermal regime in the

565 Central Andes. Pronounced spatial variability in ground temperatures, correlations with altitude and slope aspect, and distinct thermal characteristics of rock glaciers suggest that processes influencing the ground thermal regime in the Central Andes are analogous to other mountain permafrost environments. However, the unique topo-climatic and geomorphic attributes of the Andean cryosphere (including high aridity, solar radiation, lack of vegetation and organic matter, lower overall massive ice content and mountainous terrain) may enhance energy transfer with the ground compared to other permafrost regions. The

570 length of monitoring in the dataset—less than ten years of consecutive measurements—currently does not allow for assessment of long-term trends in response to climate change. In addition, the region's susceptibility to regional climate phenomena such as SAM, ENSO and PDO, which occur on decadal timescales, implies that long-term ground temperature trends for the Central Andes may only be derived from very long time-series spanning several of these cycles. The observed temporal variability in the dataset thus reflects local topographic factors and short-term microclimatic fluctuations unique to the catchment(s)

monitored.

This compilation fills a critical knowledge gap in permafrost research, providing an opportunity to refine existing permafrost distribution models in the Andean region, which were developed from indirect evidence of permafrost occurrence. Integrating insights from this study can enhance the accuracy and reliability of these models, aligning them more closely with well-calibrated models established for Europe and North America, which are based on extensive in-situ data. This study may also

support data upscaling efforts towards quantifying ground ice content in permafrost regions across broader spatial scales in the region – efforts that are essential for informing predictive models, especially in the context of climate change, where accurate assessments of permafrost dynamics are crucial for effective water resource planning and decision-making. By integrating results from this compilation, researchers can further advance the understanding of interactions between permafrost and hydrology in the Andean region under ongoing global atmospheric warming.

Many of the data collection challenges outlined in this paper are common to mountain permafrost studies elsewhere. These include natural, logistical, and financial limitations, which occasionally led to interruptions or shortened monitoring records, or introduced artifacts into the data. In addition to these challenges, this study faced constraints related to industry requirements and regulatory mandates, which dictated thermistor placement and influenced data collection and instrument maintenance schedules. Since monitoring efforts had to align with objectives of environmental impact assessments, boreholes were

established primarily to meet these needs rather than for research purposes. Consequently, instrument positioning and data collection schedules were not optimized for permafrost characterization or long-term monitoring, as would be the case in a dedicated scientific research project.

The scarcity of in-situ data in the Central Andes may reflect compounding challenges posed by difficult terrain and significant sectorial constraints. In this context, the present study represents a unique and exemplary collaboration between industry,

academia, and bi-national regulators, advancing the understanding of a key indicator of climate change in a region that is underrepresented in the Global Climate Observing System and permafrost literature. This collaboration has provided new insights into ground thermal characteristics of the Central Andes, which were not previously demonstrated with ground-based

data. It also highlights the importance of multi-stakeholder partnerships in advancing knowledge of permafrost thermal state alongside other critical issues related to climate change.

## Competing interests

At least one of the (co-)authors is a member of the editorial board of The Cryosphere. This work was partially funded by BGC Engineering Inc and University of Fribourg as a graduate research project.

## Acknowledgements

The authors would like to acknowledge the team of engineers and scientists at BGC Ingenieria Ltda. in Chile and Argentina for the collection and processing of data presented in this paper. The support from BGC Engineering Inc. and their clients in providing the opportunity to compile and analyse this dataset is further acknowledged and appreciated.

## Supplementary Information

Supplementary information related to this article is available online at ==XX URL XX==

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
