# Peer review of "Thermal State of Permafrost in the Central Andes (27°S-34°S)"

_EGUsphere, 2024_

## Author Comment (AC1)

**General Response to Reviewer 1**

We greatly appreciate the thoughtful and constructive feedback provided by Reviewer 1. We have carefully considered the suggestion to focus more on the regional baseline conditions rather than long-term trends. In response to their suggestions, we have revised the manuscript to emphasize the attribution of observed variability in ground temperatures to local climate variability. Discussions of long-term trends have been reduced and refocused to highlight the limitations of the available record length.

In addition, we have made revisions throughout the manuscript to improve clarity and conciseness, as per the reviewer's suggestions. This includes shortening sections such as the abstract and introduction, enlarging the text in Table 1, and adjusting symbols in the figures for greater clarity. We have also addressed the reviewer's detailed comments individually, which are summarized in the text below.

**Reviewer 1 Detailed Reviewer Comments & Author Responses**

L10-11 – Suggested revision "…under climate change with International collaborative efforts to collate standardized permafrost monitoring data."    (the rest of the second sentence is not necessary).

Author Response: Thank you for the comment. We have adapted the text in the abstract to limit background information, as per the reviewer's general comments. Discussion of standardized data is now only in the introduction.

L11-15 – Suggested revision "Compared to the Northern Hemisphere, there is a scarcity of ground temperature monitoring data in South America (Chile and Argentina). This has limited the understanding of thermal state and possible degradation of mountain permafrost.

Author Response: Thank you for the comment. This sentence has been revised as part of the general text reduction in the abstract. It now reads:

> *"Ground temperature monitoring data in South America are scarce compared to the Northern Hemisphere, which has historically limited the understanding of the thermal state of mountain permafrost in the region."*

L15-19 – Suggested revision "…..to compile and examine ground temperature trends for mountain (3,500 m to 5,500 m elev.) permafrost regions of the Central Andes. Ground temperature measurements from 53 boreholes along a north-south transect at the Chilean-Argentine border (27°S-34°S) reveals similarities in ground temperature characteristics with other mountain regions. This includes...."

Author Response: Adapted text accordingly.

L21-22 – Suggested revision "…observations support the hypothesis that the thermal regime of the Central…….other permafrost regions." (end of sentence is no longer required)

Author Response: Thank you for the comment. The sentence was revised to be more concise and reads:

> *"Observations support the hypothesis that the thermal regime of the Central Andes is shaped by similar processes, a perspective that was previously lacking data support."*

L22-24 – Delete sentence starting with "With the longest record.." and revise the next sentence "The high temporal variability observed in the short records (<9 years) likely reflects short-term…."

Author Response: Adapted text accordingly.

L29-31 – The last two sentences may not be necessary or at least could be combined and shorter.

> Author Response: Thank you for the comment. The text was condensed and now reads:*"This study highlights the need for ongoing ground temperature monitoring, and the critical importance of collaboration between industry, governments, and scientists to advance understanding of a key climate change indicator in a data-scarce region."*

L33-123 – The paper is not a review of global permafrost monitoring but focuses on filling a key regional gap in South America. The information on global monitoring efforts including GCOS, GTN-P etc. should be reduced. All you need to really say is that permafrost thermal state is a key indicator of permafrost change. Highlight that compared to the Northern Hemisphere where there is considerable monitoring there is a gap in South America and explain reasons why it should be addressed. Towards the end of the introduction L110-115 you describe the need for baseline permafrost information. This should come earlier as it gets lost in the lengthy discussion about global monitoring etc. You define a few acronyms in this section (e.g. GCOS, EIA) which are not used again in the paper or defined again when the terms are used. These acronyms can be deleted.

Author Response: Thank you for the comment. We have adapted the text accordingly.

L34- warming and thawing of permafrost has been documented

Author Response: Adapted text accordingly.

L45-50 – Be clear here that the active layer thickness refers to the thickness of the layer that freezes and thaws annually. This is not necessarily the same as the depth to the permafrost table particularly where permafrost is degrading and a talik has formed. The statement that the active layer is the distance from the ground surface to DZAA is incorrect as the DZAA is largely found well

below the permafrost table. It is probably better to reduce this paragraph and just give the standard definition of active layer and indicate that the permafrost temperature at DZAA (and why – filter out high frequency variation etc) is commonly used for tracking long-term change. Mentioning that the ground thermal regime and ALT are influenced by climate is fine as is mentioning that the active layer responds more to short-term variations in climate compared to deeper ground temperatures. Other information in paragraph is probably not necessary.

Author Response: Removed discussion of "thermally defined active layer" and adapted text accordingly.

L60-74 – See comments above. This section could be reduced substantially. Make the key point that there are many sites in North America, Russia/Siberia and Europe with less monitoring in central Asia, Antarctic and South America. You may want to highlight monitoring in mountain regions here (European Alps) as conditions are likely more comparable to your study.

Author Response: Thank you for the comment. We have reduced and adapted the text accordingly.

L75 "– Revise to "The lack of ground temperature data…"

Author Response: Adapted text accordingly.

L79 – revise to "…make data collection challenging (Areson…)"

Author Response: Text has been updated to increase clarity and flow. The text now reads

*"Despite a long-standing awareness of the existence of permafrost in the Andes (e.g. Catalano, 1926; Corte 1953), ground-based studies are scarce due to the region's high elevations, harsh climate and rugged terrain. Challenges such as limited funding and inadequate infrastructure for accessing remote locations further complicate data acquisition"*

L80 – suggested revision "While limited ground temperatures have been collected and analysed in the Andes (e.g. Trombotto.....), most instruments.... "

Author Response: Thank you for the comment. We have kept the text as-is because we believe it more effectively highlights the importance of the few studies that have been conducted, rather than focusing on the limited number of studies.

L89 – suggested revision "This lack of measurements in deep boreholes...:

Author Response: Adapted text accordingly.

L91 – suggested revision "....of permafrost using boreholes..."

Author Response: Adapted text accordingly.

L93 – suggested revision "Some permafrost studies in..."

Author Response: Adapted text accordingly.

L96 – revise to "..with annual increases in ALT by up to 25 cm" Note that layers have thickness rather than depth so you can refer to thickening of the active layer or deepening of the permafrost table.

Author Response: Adapted text accordingly.

L95-98 – Combine sentences and reduce text "Monnier and Kinnard (2013) installed two boreholes with thermistor strings reaching 18-25 m depth in the upper Choapa valley of northern Chile."

Author Response: Adapted text accordingly.

L100 – replace "deep" with "thick"

Author Response: Adapted text accordingly.

L100-101 – Revision suggested to combine sentences and reduce text "Preliminary data from three boreholes (20-40 m deep) installed at the Goldfields Salres Norte mining project Chile, indicated favorable conditions for permafrost between 15 and 13 m depth at one borehole (Atacama Ambiente, 2017).

Author Response: Thank you for the suggestion. We have adapted the text accordingly.

L104-105 – revision suggested "Although these other studies provide valuable...."

Author Response: Adapted text accordingly.

L110-115 – The importance of adequate baseline permafrost information for informed engineering design and environmental assessment of resource development projects should be mentioned earlier in the introduction as this is a key reason for addressing the regional permafrost knowledge gap.

Author Response: Thank for the comment. We have adapted the text to introduce this idea at the beginning of the second paragraph in Section 1.

L115 – suggested revision "Valuable data are generated that can…"

> Author Response: Thank you for the comment. We have revised the sentence for clarity and flow. The sentence now reads as: *"These investigations generate valuable data that can help assess the ground thermal regime in regions that have not yet been characterized and shared with the broader research community."*

L119-120 – suggested revision "Temperature measurements were made available…"

Author Response: Adapted text accordingly.

L121 – revision suggested "…. site conditions to support…"

Author Response: Adapted text accordingly.

L121-122 – suggested revision "All instrumentation was installed …" (EIA) can also be deleted as it is not used elsewhere in paper (where it is used you define acronym again so better to delete)

Author Response: Adapted text accordingly.

Table 1 – enlarge text

Author Response: text enlarged

L130 – Refer to Fig. 1 for location.

Author Response: Adapted text accordingly.

L137 – delete "also"

Author Response: Adapted text accordingly.

L141 – revision suggested "The eight project sites are situated at high…."

Author Response: Adapted text accordingly.

L142 – delete "also"

Author Response: Adapted text accordingly.

L156 – "remotely sensed imagery"

Author Response: Adapted text accordingly.

L157-158 – suggested revision "…and remains until spring…"

Author Response: Adapted text accordingly.

L176-195 – This section is probably too long and you could consider reducing. Given the relatively short records and the period covered by the records, you should consider what is necessary here and what might be more useful in your discussion and interpretation of results. This would reduce some of the repetition. Provide a short description of long-term change and indicate influence PDO and ENSO have in the region.

Author Response: Thank you for the comment. As suggested, we have shortened the section to describe long-term changes and reduce repetition in the discussion. We have described the influence of PDO and ENSO as reported by climate change researchers, and introduced ENSO and PDO in a more general sense earlier in the paragraph. The repeated sentence in the discussion was deleted.

L197 – Change subheading to "3.1 Ground Temperature Data and Permafrost Presence" as this is a more correct description.

Author Response: Adapted text accordingly.

L203 – Accuracy and precision of measurement system should be provided. Were data loggers used to for data collection or were manual readings made?

Author Response: We have provided the range of accuracy and precision of instruments. Most measurements were obtained with dataloggers, although occasionally manual measurements were collected. Text has been adapted to provide this detail.

L222-225 – If deep measurements below DZAA are available and temperature is below 0°C for most of the year, can you assume permafrost is present? Temperature changes are limited at these depths.

Author Response: Thank you for the comment. Yes, it is reasonable to assume permafrost is present based on these deeper measurements. Text was simplified at L222-225 and adjustments throughout the manuscript were made to refer to the sites as either "permafrost" or "non-permafrost" (i.e., "possible permafrost" was changed to "permafrost").

L256-263 – I don't think you ever described the drilling methods used. Were fluids used which would cause significant disturbance compared to methods that don't require drilling fluids.

Author response: A combination of dry sonic and diamond drilling methods were used. The latter used fluids, which could further affect early temperature measurements. A description of drilling methods was added to the text, noting the locations that could be further influenced by fluids.

L264-268 – Was the gully related to drilling disturbance or natural processes? If natural processes, the measurements were not necessarily compromised because they reflect the changes that occur at the surface, both changes in climate change and disturbance (natural or related to human activities). Knowledge of changes in surface conditions however are critical to interpretation of the ground temperature data. Assuming the casing hasn't been affected yet, the temperature data would indicate that permafrost conditions are changing in the area.

Author Response: The gully formed due to two factors: 1) A platform was constructed to install the thermistor string at a site of a previous exploration drill hole, and loose fill was used for the platform. 2) The platform altered snowmelt runoff, concentrating it at the borehole location and forming the gully. Thus, the gully resulted from both the loose fill used for the platform and the redirected runoff. This explanation has been added to the text.

L283-285 – Are these data spikes or noisy data for short periods? We have observed spikes in our data which could be electrical influences (lightning etc.). (no revision expected, just a question)

Author Response: Thank you for your question. Spikes in the data can occasionally occur for brief periods, such as those caused by electrical storms or nearby construction activities affecting ground movement. While it is not always possible to correlate these spikes with specific events to filter out noisy data, field staff were occasionally able to identify disturbances and remove the affected data when the timing of such events was clear. In cases where the cause could not be reliably determined, the noisy data were retained in the record to ensure transparency.

L325-338 – See earlier comment regarding discussion of trends. It is good you added this explanation given you are making comparison of trends determined over short periods to those in Swiss Alps where some records greater than 2 decades (same for Arctic and sub Arctic). However, is it even valid to determine trends when record is only 2-3 years long? It is important to note that for some Arctic/sub Arctic boreholes the surficial conditions are quite different from your sites. In Canada especially, there can be thick glacial sediments (including those that are fine-grained

silts/clays) with high moisture/ice contents where latent heat effects are important. Also, in circumpolar region there are frozen peatlands so thermal response will also be different from your study sites. Comparisons to mountain permafrost regions is probably better to focus on or Arctic areas where finer grained material or peatlands are less likely. Here and in Fig. 6 reference is made to cold and warm permafrost. It might be useful to indicate how you are distinguishing between these.

Author response: Thank you for the comment. We agree that it is not valid to infer long-term trends from our datasets or any record shorter than a decade. We understand that the conditions of the sites and the quality of the data are different and a direct comparison should not be done. However, we still see value in reporting estimates from other sites for reference.

The main purpose of our discussion was to highlight the significant variability within the dataset (warming vs. cooling), reflecting local topoclimatic complexity and short-term climate fluctuations. Presenting our data alongside warming rates for other regions was intended to emphasize that any long-term trends might be overshadowed by these short-term effects, including periods of cooling in the short-term in the Andes.

We also aimed to show that short-term fluctuations at non-permafrost sites are more variable than permafrost sites, likely due to factors like latent heat effects or greater thermal insulation of the ground at permafrost sites.

Additionally, we intended to convey that northern hemisphere sites show unidirectional temperature changes as climate change indicators, whereas our sites cannot yet make this claim.

We also acknowledge that thermal response in circumpolar regions will differ, as peatlands and fine-grained materials behave differently than mountain permafrost. Our goal was not to provide an in-depth comparison of thermal responses across regions, but instead to note that well-established permafrost monitoring sites in the Arctic and Alps show consistent warming based on their multidecadal datasets.

The terms "cold permafrost" and "warm/sub-arctic permafrost" are directly based on Smith et al. (2022). In their study, the authors define cold permafrost as permafrost with temperatures below -2°C. To provide additional clarity, we have included further explanations of this terminology in Section 4.1 and the caption of Figure 6.

L350 – Fig. 6 – It would be useful to give the number of sites considered in each region.

Author Response: The number of sites considered in the Andes was added to Figure 6 and Figure 10. The number of sites documenting long-term warming rates from other permafrost regions (Arctic, Sub-Arctic and Swiss Alps, aster Smith et al., 2022) were also added to Figure 6.

L354 – ALT was defined earlier so the sentence can be revised to "ALT in cryotic…". You might want to consult Riseborough (2008, ICOP) regarding using interpolation/extrapolation to determine ALT and the best approach to use.

Author Response: Thank you for the comment. We understand Riseborough's (2008, ICOP) points on extrapolating the zero-degree isotherm to estimate the active layer thickness (ALT) and agree that this may be a more optimal approach. However, we lack sufficient measurements in the active layer to confidently extrapolate temperature data from above. Additionally, snow cover varies greatly at our sites. In areas with little snow, the active layer is not thermally insulated, and atmospheric gradients affect its temperature. In contrast, where snow is present, the permafrost table is influenced by the snow cover (e.g., BTS method by Haeberli, 1978).

Our main goal is to track potential changes and compare sites, which is best achieved using a consistent approach. Thus, we focus on relative comparisons rather than absolute ALT values.

We also acknowledge linear interpolation of the zero-degree isotherm may overestimate the active layer thickness, and, due to limited shallow data and variable snow cover, it should be considered a proxy, not the exact permafrost table depth.

We have adjusted the text to convey these ideas.

L369-370 – revision suggested "…overall increase in ALT over time…"

Author Response: Thank you for the comment. This section was edited for conciseness and to limit reference to long-term trends. The text now reads:

> *"There is no consistent increase in ALT over time across the dataset, and in some locations, the active layer may be shallowing."*

L369-372 – These results reflect short-term fluctuations in climate.

Author Response: Adapted text accordingly.

L373 – What is meant by deepening of permafrost? Referring to permafrost table or base? Be clear here.

Author Response: We are referring to the permafrost table, or top of permafrost. Adapted text accordingly.

L385 – Fig. 8. Is it possible to clearly delineate talik where it occurs so that it is clear to reader?

Author Response: Added labels to Figure 8 to indicate location of taliks.

L421 – Delete ""in this dataset"

Author Response: Adapted text accordingly.

L436 Fig. 9 – some symbols are difficult to see

Author Response: Thank you for the comment. We have increased the size of symbols on the figure.

L439 – I think this section is good and makes use of the deeper temperatures that are available to give an indication of changes over several decades depending on depth.

Author Response: Thank you for the comment. No adjustments made.

L482 – suggested revision "..decrease as elevation increases (Figure...."

Author Response: Adapted text accordingly.

L487 – Be clear if permafrost depth refers to permafrost table rather than base.

Author Response: Adapted text accordingly.

L488 – suggested revision "..increase as altitude increases at..."

Author Response: Adapted text accordingly.

L510-519 – This repeats information provided in the introduction and isn't necessary. You could start the section with line 520 and be clear that your study addresses a critical knowledge gap and then describe insights revealed etc.

Author response: Thank you for the comment. The first paragraph of S 5.1. was removed since the information was stated earlier. The following text was augmented to state that the study closes a critical knowledge gap in permafrost research in the region.

L525-526 – revision suggested "..allows for characterization of average ground temperature below seasonal influences and thermal gradients...."

Author Response: Adapted text accordingly.

L534 – revision suggested "...without adequate ground temperature data..."

Author Response: Adapted text accordingly.

L562 – revision suggested "...of our sites in the Andes...."

Author Response: Adapted text accordingly.

L572-573 – Lewkowicz et al (2012) is also relevant here – uses ground surface temperature and ground temperature in the analysis to show the colder ground conditions in valley bottoms.

Author Response: Included Lewkowicz et al (2012) as a reference to the concept of temperature inversions.

L628 – You can make a stronger statement here by replacing "illustrated" with "confirmed"

Author Response: Adapted text accordingly

L632 – "from other boreholes" can be deleted (or replace with "from other sites")

Author Response: Adapted text accordingly.

L687 – (EIA) can be deleted since you give the term

Author Response: Adapted text accordingly.

L693 – Delete "(GCOS)" since you give the term

Author response: adapted text accordingly.

---

## Author Comment (AC2)

**General Response to Reviewer 2**

We would like to sincerely thank Reviewer 2 for their thoughtful and constructive feedback on our paper. In response to their general comments, we have made several revisions to improve clarity and conciseness. Specifically, we have shortened the text in the introduction and methodology sections and have relocated or condensed parts of the results section to reduce excessive qualifying language. In cases where we felt the original text provided important context, we have retained it within the results section rather than moving it to the discussion.

We agree with the reviewer that the ENSO and PDO are related but operate on different timescales. The manuscript provides a general summary of how these phenomena may affect ground temperatures, citing recent research on climate change and the Central Chile Megadrought. We also acknowledge the importance of the Southern Annular Mode (SAM) on precipitation and snow cover. Its exclusion was unintentional, and we have now added a brief description of SAM's influence in both the introduction and discussion.

We appreciate the reviewer's perspective on readability and suggestion to reorganize the results in Section 4 by choosing either site number or surface morphology, rather than both. However, we have chosen to retain both approaches, as we believe this structure is important for conveying the relationships (or lack thereof) among the data both in the context of latitude (i.e. site number) and geology/morphology. Additionally, this approach supports later discussions on the relationships with ground temperature and highlights the uniqueness of rock glaciers in the dataset. Our responses to Reviewer 2's detailed comments are provided individually, as summarized in the text below.

**Reviewer 2 Detailed Reviewer Comments & Author Responses**

Line 16: change to "(i.e., 3600 m to 5250 m)"…following the methodology section

Author Response: Thank you for the comment. The boundaries of some project areas include areas up to 5,500 m in elevation; but as you correctly point out, the boreholes are installed only as high as ~5,250 m. Text revised accordingly.

Line 17-18: I didn't see any borehole at 34° S. I suggest adjusting the latitude range to 27° - 33° S

Author Response: Thank you for the comment. The property boundaries of some of the projects extend to 34° S. However, the location of projects shown on Figure 1 represent the approximate centres of project boundaries, and not necessarily positions of the boreholes. While we are unable to share the exact coordinates of the boreholes, some of them are indeed located closer to 34° S, so it is most accurate to keep the boundaries as presented.

Line 69-72: could be just "Despite…. monitoring boreholes". In a different paragraph, you may detail why the "challenges remain"

Author Response: Thank you for the comment. We revised this paragraph in response to Reviewer 1's comments on L60-74 to make it more concise. However, we believe it's still important to first mention the common challenges faced in the northern hemisphere before transitioning to the specific issues in the Andes. Therefore, we have retained the mention of challenges at northern hemisphere sites in the current paragraph. The revised text now reads:

> *"However, assessing long-term changes in response to climate warming remains challenging at many locations due to sparse borehole distribution and short of discontinuous monitoring records. These challenges are often the result of site access difficulties, high installation and maintenance costs and the narrow focus of monitoring programs, particularly in mountain permafrost regions."*

Line 79: delete "Which make data extremely challenging to obtain"

Author Response: Please see our response to Reviewer #1's comment on this line. We have kept this part of the sentence, but revised to increase clarity. The text now reads:

*"Despite a long-standing awareness of the existence of permafrost in the Andes (e.g. Catalano, 1926; Corte 1953), ground-based studies are scarce due to the region's high elevations, harsh climate and rugged terrain. Challenges such as limited funding and inadequate infrastructure for accessing remote locations further complicate data acquisition"*

Line 84-86: you state two ideas in one paragraph "Consequently...below DZAA" and "limiting...is evident". Rephrase

Author Response: Thank you for the comment. We have revised the sentence to increase clarity. The sentence now reads as

> *"Consequently, monitoring records often lack the duration and depth required to discern average ground temperatures or trends in warming/cooling below the DZAA, limiting comparisons of the ground thermal state in the Andes with regions where permafrost degradation is evident."*

Line 103-104: could be just "The preliminary...5 m and 13 m depth at one location". The second part of your paragraph does not contribute at all

Author Response: Text revised accordingly.

Line 112-113: unnecessary details. The focus isn't on mining. You could be more general

Author Response: Thank you for the comment. We believe it is important to highlight what is currently driving data collection in the region and how research advancements may be facilitated through collaboration with parties already engaged in monitoring efforts. In many cases, this

involves mining, presenting an opportunity for productive industry partnerships. It is our opinion that it is appropriate to retain the original text, as it does not specifically mention mining, but rather the broader context of precious metals and natural resource development.

Line 113: "Site investigation" or "Minning sites". What is the focus of this paragraph?

Author Response: Thank you for the comment. The paragraph refers to general environmental assessments needed for natural resource development, not specifically mining. To clarify, we've revised the text to:

> "Scientific investigations, often including the collection of ground temperature data in permafrost zones, are necessary to support environmental permitting and engineering designs."

Line 117: "Semi-arid Central Andes"? to be consistent with the concepts. Is Central Andes or Semi-arid Central the central term?

Author Response: Thank you for the comment. While it is correct to use the qualifying term "semi-arid" it is not necessary in the context of this paragraph. Text has been updated accordingly.

In Fig 1: I wonder if A and D are necessary for the legend

Author Response: We agree that A and D are not necessary in the legend as neither equatorial or polar climates appear on Figure 1. Legend was updated accordingly.

In Fig 1: It would be great to know the year and authors who updated the Koppen-Geiger climate classification

Author Response: Koppen-Geiger zones are based om Kottek et al., (2006) for Argentina and Sarricolea et al., (2017) for Chile. We have updated the figure caption and added the citations to the list of references, which are:

> Kottek, M., J. Grieser, C. Beck, B. Rudolf, and F. Rubel, 2006: World Map of the Köppen-Geiger climate classification updated. Meteorol. Z., 15, 259-263. DOI: 10.1127/0941-2948/2006/0130.

> Sarricolea, P., Herrera-Ossandon, M. & Meseguer-Ruiz, Ó. (2017). Climatic regionalisation of continental Chile. *Journal of Maps, 13*(2), 66–73. https://doi.org/10.1080/17445647.2016.1259592.

Line 136-137: 2,000 mm as average annual precipitation is extremely high for the latitudes of your study site. Please check

Author Response: Thank you for the comment. The stated value of 2,000 m was incorrect and citation was incomplete. We have corrected the value to 500 mm and added the source Viale and Garreaud (2015)

> Viale, M. and Garreaud, R. (2015) Orographic effects of the subtropical and extratropical Andes on upwind precipitating clouds. Journal of Geophysical Research: Atmospheres, 120, 4962–4974.

Line 205: Then, how was the frequency of data collection expressed in the results? monthly average? Seasonal average?

Author Response: The data are generally expressed as monthly averages (e.g., Figure 5) and, in some cases, as annual averages or most recent measurements when sufficient data for averages were not available (e.g., Figures 9, 11, 12, 13). The supplementary information package includes raw data in time series, reflecting the frequency at which it was collected (e.g., hourly, daily).

Line 148-152: could be simpler. For instance: "Seasonal average temperatures at 4000 m range from XXX XXx in July (reference). Nevertheless, some rock extend at 3500 m (reference)"

Author Response: Thank you for the comment. We have simplified the text, which now reads as:

> "With seasonal average temperatures ranging from ~18°C in January to ~8°C in July, permafrost will not form and is unlikely to exist in this zone; where it does exist, it is naturally degrading."

Line 153-154: is contradictory with the previous sentence

Author Response: Thank you for the comment. We aim to convey that summer temperatures are above 0°C, while annual temperatures are not. To improve clarity, we have revised the text as follows:

> "In contrast, the climate of the ET belt (elevations generally >4,000 m) is associated with low temperatures year-round and mean annual air temperatures (MAATs) below 0°C (Vuille et al., 2003; Garreaud, 2009), creating favourable conditions for permafrost formation. During the summer months (December to February), temperatures within the ET belt remain above 0°C, with daily highs exceeding 15°C and averages below 10°C. "

Line 162-164: It sounds as discussion. Why talk about other mountain regions in your study area section?

Author Response: Thank you for the comment. We have adjusted the text and cited literature to be specific to the Andes. The text now reads as:

*"Slight variations in altitude (usually spanning a few hundred metres) can generate marked differences in air temperature, precipitation, vegetation, snowpack, solar radiation, and glacial cover over short lateral distances (Arenson et al., 2022)."*

Line 164-169: this could be just one phrase. Once you have presented all the topographic and climate background, you may state "topo-climate conditions in Central Andes favors a highly heterogeneous occurrence of permafrost in the region (Hilbich et al., 2022)"

Author Response: Thank you for the comment. These second sentence discusses the specific subsurface processes (i.e., infiltration, freeze/thaw and movement of air and water) influencing the heterogeneity of permafrost, which are induced by topo-climatic conditions. We have decided to retain the original text as it accurately reflects this distinction.

Line 173-176: Repeated. You already state that permafrost has heterogeneous occurrence

Author Response: Thank you for the comment. These lines discuss the distribution of permafrost in relation to slope orientation, not just the heterogenous occurrence of permafrost. We have decided to retain the original text as it accurately reflects this distinction.

Line 173: I suggest "the southern margin of our study area (32-33°S)…"

Author Response: Please see our response to comments on Line 17-18. It is most accurate to keep the southern boundary at 34° S.

Line 177-178: which several studies? Please, cite them

Author Response: Thank you for the comment. We have listed the studies at the end of the sentence.

Line 194-195: it is not necessary. I suggest deleting

Author Response: Thank you for the comment. Since Section 2 aims to describe the regional setting in relation to the permafrost thermal regime, we have decided to retain the reference. This text is relevant as it discusses climate and infiltration processes that directly influence the ground's thermal regime. However, in response to Reviewer 1's comment on Lines 176-195, the section has been shortened and made more concise.

Line 198-200: Re-phrase to something like "Ground temperature was monitored between the years 2006 and 2017 at eight sites located distributed near the Chile and Argentinean border (Fig.1)"

Author Response: Thank you for your comment. We aimed to clarify that the monitoring start dates varied within the given timeframe and did not all begin in 2006. The data for some locations extend to 2022 and continue to the present, with some locations still actively collecting data. We have revised the text as follows:

*"Between 2006 and 2017, ground temperature monitoring was systematically initiated at eight industrial project sites located between 27°S and 34°S and within approximately 25 km of the Chile-Argentine border (Figure 1)."*

Line 211-214: Re-phrase to something like "The duration of monitoring varies from less than a month to nine years. Several monitories had interruptions that were attributed to climate/terrain factors (i.e., slope instability) and lack manutention". After this phrase, you must explain how to solve the gaps

Author Response: Thank you for the comment. We have updated the text as follows to improve clarity:

*"The duration of monitoring varied by borehole, ranging from less than one month to nine years (Figure 2). Several locations experienced interruptions to data collection due to factors such as electrical storms, instrument malfunctions, inaccessibility for download or maintenance due to remoteness, adverse weather, slope instability and/or changing regulatory requirements."*

We discuss how the gaps were solved in Section 3.2.

Line 214-216: It is not necessary. Perhaps keep "A comprehensive discussion of data collection is included in section 3.2"

Author Response: We appreciate the comment, but think it is important to introduce the challenges of data collection in a general sense, before going into detail in Section 3.2. We have therefore decided to keep the text as-is.

Line 218-220: It is not necessary. I suggest deleting

Author Response: Thank you for the comment, which we addressed in combination with a similar comment by reviewer 1 (RC1 Comment L222-L225). As a result, we have significantly reduced the discussion of permafrost presence in this part.

Line 227-228: Needs to be relocated after you describe the existence of "gaps" (i.e., line 211-214)

Author Response: Thank you for the comment. This is addressed as part of our response to comments on Lines 211-214.

Line 235: It is not necessary. I suggest deleting

Author Response: Adapted text accordingly.

Line 238-241: must be shortened

Author Response: Thank you for the comment, text was simplified and now reads as

*"Challenging terrain, remoteness, and lack of financial support can limit site access, leading to poor instrument maintenance and data loss. Other factors affecting ground temperature monitoring unrelated to mountainous regions may include instrument damage by wildlife, vandalism, construction, or damage during instrument installation."*

Line 242-251: must be shortened

Author Response: Thank you for the comment. The text was simplified and now reads as:

*"Many of these challenges were encountered across the study area, with most interferences or malfunctions identified by field staff before analysis. The most common causes of data interruptions were battery power loss, faulty connections, or sensor failures between maintenance visits. Irregular funding for ground temperature monitoring based on specific project needs, led to inconsistent and unpredictable field work and maintenance schedules, making it challenging to perform routine upkeep (e.g., battery replacement of sensor repairs) proactively or in a timely manner."*

Line 255-259: It sounds like results. Relocate

Author Response: Thank you for the comment. We believe this information is appropriate for its current placement, as it describes the preparation of datasets and the rationale for discarding anomalous data before interpretation. We therefore propose to keep the text as-is.

Line 259-262: re-write to something like "Boreholes affected by drilling disturbances had less weight in the analyses". What do you mean with "weight"? clarify

Author Response: Thank you for the comment. We meant that these measurements were considered with caution, without applying a formal weighting scheme. The text now reads:

*"Early measurements from Site 1 and other locations with approximately one year of data (i.e., boreholes 6 8 and 6-11, 7-4, 7-5 and 7-6) were treated with caution in the analyses, particularly those collected during the initial two to three months , as they are unlikely to represent average ground temperature conditions."*

Line 265-266: It is not necessary. I suggest deleting

Author Response: Thank you for the comment. We believe that discussing the changing landscape is integral to understanding monitoring conditions in mountain permafrost, as well as the communicating the unique challenges posed by industrial sites. These topics are of interest to the permafrost community, and therefore, we propose to retain these lines in the paper.

Line 266-270: It is part of the discussion

Author Response: Thank you for the comment. While we understand the reviewer's perspective, we believe that this section is more appropriately placed in the results, as it explains the rationale for discarding anomalous data prior to interpretation. Therefore, we propose to retain the text in place.

Line 275-286: It is part of results and/or discussion. You could erase these lines

Author Response: Similar to our response to feedback on Line 266-270, we propose to keep the text in its current placement as it describes site conditions and rationale around data processing.

Line 301: replace "goodness" by another term or re-write

Author Response: Replaced "goodness" with "quality".

Line 323-324: why does this reference to northern permafrost appear here? Move to discussion

Author Response: Thank you for the comment. The reference to the northern hemisphere is included to highlight the extensive research on permafrost warming and to provide a reference for our results. This reference helps contextualize our dataset within the broader body of permafrost research, which has clearly demonstrated the link between permafrost degradation and global climate change.

Our data from the Andes, on the other hand, do not (yet) show similar warming patterns, and a detailed analysis of warming trends or comparisons with northern hemisphere regions is not feasible in the discussion section of the paper due to the limited duration of our dataset.

Therefore, we propose to retain the text as-is in its current location, as it serves only to contextualize our findings without further discussion in the paper.

Line 330-342: Again, why do you refer to other permafrost regions in your results? Move to the discussion in a subsection called something like "thermal state of permafrost regions"

Author Response: Thank you for your comment. As mentioned in our response to the comment on Lines 323-324, we reference other permafrost regions to help contextualize our results within the broader body of permafrost research, without intending to discuss them in detail. Therefore, we suggest keeping the text in its current location.

In Fig 5 and 7: What do you mean with "Glacial Deposit"? moraines? are glacial deposits cryotic or non-cryotic features? Besides, please check the "@" in the label

Author Response: By Glacial Deposits we mean sediments left behind by a moving glacier in general, and this isn't just limited to moraines. While glacial deposits can sometimes be cryotic features, in this dataset they are non-cryotic. We believe the "@" symbol in the label is a display error caused by the download process, and as such, we have not made any changes to the figures.

Fig 6: Move to discussion. It is valuable data. But not in your results

Author Response. Please refer to our response to the comment on Lines 323-324. Figure 6 is intended to serve as a point of reference of other studies for context only. Since we are not yet able to evaluate long term trends in the Andes data, we feel it is best to retain Figure 6 here alongside the results rather than being moved to the discussion section.

Line 354-355: why is this line not in the section methodology?

Author Response: As with Sections 4.1, 4.3, and 4.4, section 4.2 includes a brief description on how the plotted parameter was calculated. To streamline the text and avoid repetition, we opted not to include these details in the methodology. We therefore propose keeping the text as-is.

Line 367-369: It is not necessary. I suggest deleting

Author Response: Thank you for your comment. The high depth and variability observed in our boreholes within rock glaciers is not typical of rock glaciers in general—many rock glaciers in the Alps, for example, show relatively shallow permafrost depths. Additionally, this characteristic may not apply to all rock glaciers in the Andes, and our observation may be influenced by sampling location bias.

Since the observed depth variability is likely specific to the sites visited and not a general characteristic of rock glaciers, we prefer to retain the statement because it reflects the advanced degradation noted by the field team.

Line 369-371: Move to discussion ("Similar...atmospheric warming")

Author Response: Thank you for the comment. We have shortened the sentence to avoid referencing long-term trends, but have not expanded on the idea in the discussion section. As noted in our response to the comment on Lines 323-324, we acknowledge it is not feasible to analyze long-term trends due to the limited duration of our dataset. We have also made clarifications throughout the paper to address this more clearly, which is why the idea has not been moved to the discussion.

Line 395-400: Move to discussion

Author Response: Thank you for the comment. We believe that the introductory sentences describing factors influencing profile shape provide helpful context for the results and are more appropriately placed in this section rather than in the discussion. We have edited the text to make it more concise. The text now reads:

> *"The temperature profile within a borehole is shaped by factors such as regional geothermal heat flux, lithology variations, local topo-climatic variations, and historical fluctuations in ground surface temperatures. These factors result in a wide range in ground temperatures (from approximately -7 to 7°C) and varied profile curvatures (Figure 9), reflecting the complex thermal landscape of the Andes."*

Line 402: I think section 4.3. starts here

Thank you for the comment. Please see our response to comments on Lines 395-400. The text has been edited for conciseness, but the introductory ideas have been retained for context.

Line 412-419: Move to discussion

Thank you for the comment. The text in this section has been edited and reduced for clarity, focusing on results and minimal interpretation.

Line 421-430: Move to discussion or present just as results

Thank you for the comment. The text has been edited in this section to present results only, with minimal interpretation.

Line 441: delete "may" and "some"

Author Response: Thank you for the suggestions, We have adapted the text accordingly.

Line 443-447: It sounds like methods. Move or delete

Author Response: Please see our response to comment on Line 354-355. A brief description of how the plotted parameter was calculated has been included here. To streamline the text and avoid repetition, we chose not to include these details in the methodology. We propose keeping the text as-is.

Line 468-472: Move to discussion or conclusion

Thank you for your comment. These lines are intended to describe the results presented in the figures, and we feel that this is best conveyed in this section rather than the discussion. Since the text focuses solely on the presentation of the results, without elaborating on their implications, we suggest leaving the text as is.

Line 476-478: Move to discussion

Thank you for your comment. In these lines, we are summarizing the results of the compilation and providing a brief remark on the uniqueness of rock glaciers. As mentioned in our previous response, we believe this is best presented in this section rather than the discussion, and we suggest leaving the text as is.

Line 495: move to the discussion

Thank you for the comment. The text has been shortened in this section to present results only, with minimal interpretation.

Line 532-534: re-write. I suggest something like this "This work provides temperature and depth permafrost data recovered from in situ measurements, evidencing XXX. Our findings contrast (or are coherent) with statistical methods…"

Author Response: Thank you for the comment. We have simplified the text; however, we have not discussed specific implications for permafrost distribution models developed by others. Our primary aim with this idea was to emphasize that in-situ data are now available to update or develop new models, whereas previously models were based solely on field observations and statistical methods used by other researchers. Our text now reads as:

> "Another important implication of this work is that the data can be used to validate existing permafrost distribution models in the region, which were previously developed without any borehole temperature data (e.g., Arenson and Jakob, 2010; Ruiz and Trombotto, 2012). Additionally, having ground temperature data within permafrost and non-permafrost zones helps reduce the risk of site-selection bias towards permafrost presence, which can complicate evaluations of spatially distributed models predicting permafrost."

Line 553: "medium"? Do you mean ground properties?

Author Response: Yes; we have adapted the text accordingly.

Line 579: Please check the reference of Schultz et al., 2012. That paper refers to the Arid Coastal of Chile (18-30° S)

Author Response: Thank you for the comment. This sentence has been removed in response to a comment from Reviewer #1 on L176-195 to reduce repetition. It is correct however that this paper refers to coastal Chile. We have indicated this in the text within Section 2, where the reference to the paper is made.

Line 593: would be great to recall the latitude when you cite Site 3

Author Response: Thank you for the suggestion. We have adjusted the text accordingly.

---

## Author Response (AR1)

Dear Editor,

We would like to sincerely thank you for your thoughtful feedback, which has greatly improved the quality of our paper. We appreciate the time taken to review our responses to the reviewers' comments, as well as your additional suggestions and guidance. In response, we have further revised the manuscript to address the points raised and made editorial changes to improve both clarity and flow.

In response to your specific request regarding other studies, we have followed your guidance to further clarify which information was from previous studies in order to retain Figure 6 in place. Specifically, the main text now reads:

> *" Figure 6 includes representative long-term warming trends from other permafrost regions as documented in Smith et al. (2022). This includes continuous or "cold" Arctic permafrost (temperatures below -2°C, warming rates from ~0.04 to 0.11 °C/yr, monitored since the 1980s), discontinuous or "warm" Arctic permafrost (temperatures between -2° and 0°C, warming rates from~0.01 to 0.05°C /yr, monitored since the late 1970s to early 1980s) and mountain permafrost within the Swiss Alps (average: ~0.02 °C/yr, monitored since in the late 1980s to early 1990s)."*

We have also modified the figure caption to provide this added clarity, which now reads:

> *"Figure 1: Short-term ground temperature changes with time at 20 m depth in the Andes (this study), shown alongside long-term ground temperature trends compiled by Smith et al., (2022) in permafrost regions of the northern hemisphere (i.e., Cold Arctic, Warm / Sub-Arctic and Swiss Alps). The Andean estimates (n permafrost sites = 19; n non-permafrost sites = 9) are based on 2-9 years' worth of measurement and reflect short-term fluctuations in climate. Warming rates for other studies (n Cold Arctic Sites = 9; n Warm / Sub-Arctic Sites = 11; n Swiss Alps sites = 5) were derived from decadal / multi-decadal datasets and represent the effects of climate change. The terms "Cold Arctic permafrost" and "Warm/Sub-Arctic permafrost," from Smith et al. (2022), distinguish cold permafrost as below -2°C and warm permafrost as closer to 0°C. Estimates for the Andes sites are summarized in Table S1 with corresponding $r^2$ values."*

In response to your request for more specificity regarding changes made in response to Reviewer #1, we have included an updated point-by-point reply at the end of this document, with additions tracked in red text. Furthermore, we have provided an updated version of the manuscript with all reviewer suggestions and changes implemented.

Once again, we thank you and both reviewers for your invaluable feedback, which has significantly enhanced the quality of our manuscript. We appreciate your input and hope that these revisions meet your expectations.

**Point-by-point Responses to Reviewer #1 with Detailed Text Adaptations**

L10-11 – Suggested revision "...under climate change with International collaborative efforts to collate standardized permafrost monitoring data."    (the rest of the second sentence is not necessary).

Author Response: Thank you for the comment. We have adapted the text in the abstract to limit background information, as per the reviewer's general comments. Discussion of standardized data is now only in the introduction.

L11-15 – Suggested revision "Compared to the Northern Hemisphere, there is a scarcity of ground temperature monitoring data in South America (Chile and Argentina). This has limited the understanding of thermal state and possible degradation of mountain permafrost.

Author Response: Thank you for the comment. This sentence has been revised as part of the general text reduction in the abstract. It now reads:

> *"A paucity of ground temperature data in South America has historically limited the characterization of the thermal state of permafrost in the Andes compared to other mountain regions."*

L15-19 – Suggested revision ".....to compile and examine ground temperature trends for mountain (3,500 m to 5,500 m elev.) permafrost regions of the Central Andes. Ground temperature measurements from 53 boreholes along a north-south transect at the Chilean-Argentine border (27°S-34°S) reveals similarities in ground temperature characteristics with other mountain regions. This includes...."

Author Response: Thank you for the comment. This text has been revised to be more concise and now reads:

> *"This study represents the first coordinated effort to compile and examine regional baseline conditions using ground temperature data from mountain permafrost of the Central Andes (3,500 m to 5,250 m, 27°S-34°S). Measurements from 53 boreholes along a north-south transect at the Chilean-Argentine border reveal ground temperature characteristics similar to other mountain permafrost regions, including high spatial and temporal variability, correlations with altitude and slope aspect, and distinct thermal attributes of rock glaciers"*.

L21-22 – Suggested revision "...observations support the hypothesis that the thermal regime of the Central.......other permafrost regions." (end of sentence is no longer required)

Author Response: Thank you for the comment. The sentence was revised to be more concise and reads:

*"Observations support the hypothesis that the thermal regime of the Central Andes is shaped by similar processes, a perspective that was previously lacking data support."*

L22-24 – Delete sentence starting with "With the longest record.." and revise the next sentence "The high temporal variability observed in the short records (<9 years) likely reflects short-term...."

Author Response: Thank you for the comment. The sentence now reads:

*"The high temporal variability observed in the short records (<9 years) reflects short-term microclimatic fluctuations and topo-climatic attributes unique to the Andean cryosphere."*

L29-31 – The last two sentences may not be necessary or at least could be combined and shorter.

Author Response: Thank you for the comment. The text was condensed and now reads:*"This study highlights the need for ongoing ground temperature monitoring, and the critical importance of collaboration between industry, governments, and scientists to advance understanding of a key climate change indicator in a data-scarce region."*

L33-123 – The paper is not a review of global permafrost monitoring but focuses on filling a key regional gap in South America. The information on global monitoring efforts including GCOS, GTN-P etc. should be reduced. All you need to really say is that permafrost thermal state is a key indicator of permafrost change. Highlight that compared to the Northern Hemisphere where there is considerable monitoring there is a gap in South America and explain reasons why it should be addressed. Towards the end of the introduction L110-115 you describe the need for baseline permafrost information. This should come earlier as it gets lost in the lengthy discussion about global monitoring etc. You define a few acronyms in this section (e.g. GCOS, EIA) which are not used again in the paper or defined again when the terms are used. These acronyms can be deleted.

Author Response: Thank you for the comment. We have removed the discussion about global monitoring efforts and refocused the paragraph on the regional gap in South America and importance of addressing it. We also removed GCOS, GTN-P and EIA acronyms. The need for baseline monitoring data was mentioned earlier in the paragraph as suggested.

L34- warming and thawing of permafrost has been documented

Author Response: Thank you for the comment. The sentence now reads:

*"Several studies have documented large-scale warming and thawing of permafrost in recent decades"*

L45-50 – Be clear here that the active layer thickness refers to the thickness of the layer that freezes and thaws annually. This is not necessarily the same as the depth to the permafrost table particularly where permafrost is degrading and a talik has formed. The statement that the active

layer is the distance from the ground surface to DZAA is incorrect as the DZAA is largely found well below the permafrost table. It is probably better to reduce this paragraph and just give the standard definition of active layer and indicate that the permafrost temperature at DZAA (and why – filter out high frequency variation etc) is commonly used for tracking long-term change. Mentioning that the ground thermal regime and ALT are influenced by climate is fine as is mentioning that the active layer responds more to short-term variations in climate compared to deeper ground temperatures. Other information in paragraph is probably not necessary.

Author Response: We have removed the discussion of the 'thermally defined active layer' and streamlined the paragraph accordingly. The revised version now focuses on the standard definition of the active layer and highlights that permafrost temperature is measured at DZAA because it lies below the influence of seasonal variations, making it ideal for monitoring long-term changes. Additionally, we clarified the distinction between the active layer thickness and the depth to the permafrost table.

L60-74 – See comments above. This section could be reduced substantially. Make the key point that there are many sites in North America, Russia/Siberia and Europe with less monitoring in central Asia, Antarctic and South America. You may want to highlight monitoring in mountain regions here (European Alps) as conditions are likely more comparable to your study.

Author Response: Thank you for the comment. We have reduced and adapted the text. The paragraph now only briefly mentions the many sites in the northern hemisphere with less in Central Asia and the South.

L75 "– Revise to "The lack of ground temperature data…"

Author Response: Thank you for the comment. The sentence now reads:

*"The lack of ground temperature data, especially at depths greater than a few metres, is particularly pronounced in South America, where permafrost regions are largely understudied compared to other regions of the world"*

L79 – revise to "…make data collection challenging (Areson…)"

Author Response: Text has been updated to increase clarity and flow. The text now reads

*"Despite a long-standing awareness of the existence of permafrost in the Andes (e.g. Catalano, 1926; Corte 1953), ground-based studies are scarce due to the region's high elevations, harsh climate and rugged terrain. Challenges such as limited funding and inadequate infrastructure for accessing remote locations further complicate data acquisition"*

L80 – suggested revision "While limited ground temperatures have been collected and analysed in the Andes (e.g. Trombotto.....), most instruments.... "

Author Response: Thank you for the comment. We have kept the text as-is because we believe it more effectively highlights the importance of the few studies that have been conducted, rather than focusing on the limited number of studies.

L89 – suggested revision "This lack of measurements in deep boreholes...:

Author Response: Thank you for the suggestion. We have revised the text as follows:

*"This lack of deeper measurements has been acknowledged in a proposed permafrost national monitoring plan for Chile..."*

L91 – suggested revision "....of permafrost using boreholes..."

Author Response: Thank you for the comment: The text now reads:

*"....which recommends long-term monitoring using boreholes extending to the base of permafrost."*

L93 – suggested revision "Some permafrost studies in..."

Author Response: Thank you for the comment: The text now reads:

*"Some permafrost studies in....."*

L96 – revise to "..with annual increases in ALT by up to 25 cm" Note that layers have thickness rather than depth so you can refer to thickening of the active layer or deepening of the permafrost table.

Author Response: Thank you for the comment. The text now reads:

*"....with annual increases in ALT by up to 25 cm."*

We have also revised text in other areas of the document to refer to thickening of the active layer or deepening of the permafrost table.

L95-98 – Combine sentences and reduce text "Monnier and Kinnard (2013) installed two boreholes with thermistor strings reaching 18-25 m depth in the upper Choapa valley of northern Chile."

Author Response: Thank you for the comment. The sentences were combined and the text now reads:

*"Monnier and Kinnard (2013) reported findings from two boreholes with thermistor strings reaching 18-25 m in the upper Choapa valley of northern Chile."*

L100 – replace "deep" with "thick"

Author Response: Replaced "deep" with "thick".

L100-101 – Revision suggested to combine sentences and reduce text "Preliminary data from three boreholes (20-40 m deep) installed at the Goldfields Salres Norte mining project Chile, indicated favorable conditions for permafrost between 15 and 13 m depth at one borehole (Atacama Ambiente, 2017).

Author Response: Thank you for the suggestion. The text now reads:

*"Preliminary data from three boreholes (20-40 m deep) installed at the Goldfields Salares Norte mining project in Chile, indicated favorable conditions for the presence of permafrost between approximately 5 m and 13 m depth at one borehole (Atacama Ambiente, 2017)."*

L104-105 – revision suggested "Although these other studies provide valuable...."

Author Response: Thank you for the suggestion, The text now reads:

*"Although these studies provide valuable insights into the thermal state and changes to permafrost....."*

L110-115 – The importance of adequate baseline permafrost information for informed engineering design and environmental assessment of resource development projects should be mentioned earlier in the introduction as this is a key reason for addressing the regional permafrost knowledge gap.

Author Response: Thank for the comment. We have adapted the text to introduce this idea at the beginning of the second paragraph in Section 1.

L115 – suggested revision "Valuable data are generated that can..."

Author Response: Thank you for the comment. We have revised the sentence for clarity and flow. The sentence now reads as: *"These investigations generate valuable data that can help assess the ground thermal regime in regions that have not yet been characterized and shared with the broader research community."*

L119-120 – suggested revision "Temperature measurements were made available..."

Author Response: Thank you for the comment. The text now reads:

*"Temperature measurements were made available to the authors by BGC Engineering Inc., with permission of the individual project owners."*

L121 – revision suggested "…. site conditions to support…"

Author Response: Thank you for the comment. The text now reads:

*"The data were accompanied by confidential field notes and reports detailing instrumentation and site conditions to support the interpretations presented."*

L121-122 – suggested revision "All instrumentation was installed …" (EIA) can also be deleted as it is not used elsewhere in paper (where it is used you define acronym again so better to delete)

Author Response: Deleted EIA as suggested.

Table 1 – enlarge text

Author Response: text enlarged

L130 – Refer to Fig. 1 for location.

Author Response: Thank you for the comment. Reference to Figure 1 was added.

L137 – delete "also"

Author Response: "also" deleted.

L141 – revision suggested "The eight project sites are situated at high…."

Author Response: Thank you for the comment. The text now reads:

*"The eight project sites in this study are situated at high altitudes,"*

L142 – delete "also"

Author Response: "also" deleted.

L156 – "remotely sensed imagery"

Author Response: Thank you for the comment. The text now reads:

*"Surface geomorphic indicators of permafrost in this belt, identified in the field and through remotely sensed imagery, include rock glaciers, gelifluction slopes, and patterned ground"*

L157-158 – suggested revision "…and remains until spring…"

Author Response: Thank you for the comment. The text now reads:

*"In both climatic belts, snow accumulates at high altitudes during winter and remains until spring."*

L176-195 – This section is probably too long and you could consider reducing. Given the relatively short records and the period covered by the records, you should consider what is necessary here and what might be more useful in your discussion and interpretation of results. This would reduce some of the repetition. Provide a short description of long-term change and indicate influence PDO and ENSO have in the region.

Author Response: Thank you for the comment. As suggested, we have shortened the section to describe long-term changes and reduce repetition in the discussion. We have described the influence of PDO and ENSO as reported by climate change researchers, and introduced ENSO and PDO in a more general sense earlier in the paragraph. The repeated sentence in the discussion was deleted.

L197 – Change subheading to "3.1 Ground Temperature Data and Permafrost Presence" as this is a more correct description.

Author Response: Subheading changed to "3.1 Ground Temperature Data and Permafrost Presence".

L203 – Accuracy and precision of measurement system should be provided. Were data loggers used to for data collection or were manual readings made?

Author Response: We have provided the range of accuracy and precision of instruments. Most measurements were obtained with dataloggers, although occasionally manual measurements were collected. Text has been adapted to provide this detail.

L222-225 – If deep measurements below DZAA are available and temperature is below 0°C for most of the year, can you assume permafrost is present? Temperature changes are limited at these depths.

Author Response: Thank you for the comment. Yes, it is reasonable to assume permafrost is present based on these deeper measurements. Text was simplified at L222-225 and adjustments

throughout the manuscript were made to refer to the sites as either "permafrost" or "non-permafrost" (i.e., "possible permafrost" was changed to "permafrost").

L256-263 – I don't think you ever described the drilling methods used. Were fluids used which would cause significant disturbance compared to methods that don't require drilling fluids.

Author response: A combination of dry sonic and diamond drilling methods were used. The latter used fluids, which could further affect early temperature measurements. A description of drilling methods was added to the text, noting the locations that could be further influenced by fluids.

L264-268 – Was the gully related to drilling disturbance or natural processes? If natural processes, the measurements were not necessarily compromised because they reflect the changes that occur at the surface, both changes in climate change and disturbance (natural or related to human activities). Knowledge of changes in surface conditions however are critical to interpretation of the ground temperature data. Assuming the casing hasn't been affected yet, the temperature data would indicate that permafrost conditions are changing in the area.

Author Response: The gully formed due to two factors: 1) A platform was constructed to install the thermistor string at a site of a previous exploration drill hole, and loose fill was used for the platform. 2) The platform altered snowmelt runoff, concentrating it at the borehole location and forming the gully. Thus, the gully resulted from both the loose fill used for the platform and the redirected runoff. This explanation has been added to the text.

L283-285 – Are these data spikes or noisy data for short periods? We have observed spikes in our data which could be electrical influences (lightning etc.). (no revision expected, just a question)

Author Response: Thank you for your question. Spikes in the data can occasionally occur for brief periods, such as those caused by electrical storms or nearby construction activities affecting ground movement. While it is not always possible to correlate these spikes with specific events to filter out noisy data, field staff were occasionally able to identify disturbances and remove the affected data when the timing of such events was clear. In cases where the cause could not be reliably determined, the noisy data were retained in the record to ensure transparency.

L325-338 – See earlier comment regarding discussion of trends. It is good you added this explanation given you are making comparison of trends determined over short periods to those in Swiss Alps where some records greater than 2 decades (same for Arctic and sub Arctic). However, is it even valid to determine trends when record is only 2-3 years long? It is important to note that for some Arctic/sub Arctic boreholes the surficial conditions are quite different from your sites. In Canada especially, there can be thick glacial sediments (including those that are fine-grained silts/clays) with high moisture/ice contents where latent heat effects are important. Also, in circumpolar region there are frozen peatlands so thermal response will also be different from your

study sites. Comparisons to mountain permafrost regions is probably better to focus on or Arctic areas where finer grained material or peatlands are less likely. Here and in Fig. 6 reference is made to cold and warm permafrost. It might be useful to indicate how you are distinguishing between these.

Author response: Thank you for the comment. We agree that it is not valid to infer long-term trends from our datasets or any record shorter than a decade. We understand that the conditions of the sites and the quality of the data are different and a direct comparison should not be done. However, we still see value in reporting estimates from other sites for reference.

The main purpose of our discussion was to highlight the significant variability within the dataset (warming vs. cooling), reflecting local topoclimatic complexity and short-term climate fluctuations. Presenting our data alongside warming rates for other regions was intended to emphasize that any long-term trends might be overshadowed by these short-term effects, including periods of cooling in the short-term in the Andes.

We also aimed to show that short-term fluctuations at non-permafrost sites are more variable than permafrost sites, likely due to factors like latent heat effects or greater thermal insulation of the ground at permafrost sites.

Additionally, we intended to convey that northern hemisphere sites show unidirectional temperature changes as climate change indicators, whereas our sites cannot yet make this claim.

We also acknowledge that thermal response in circumpolar regions will differ, as peatlands and fine-grained materials behave differently than mountain permafrost. Our goal was not to provide an in-depth comparison of thermal responses across regions, but instead to note that well-established permafrost monitoring sites in the Arctic and Alps show consistent warming based on their multidecadal datasets.

The terms "cold permafrost" and "warm/sub-arctic permafrost" are directly based on Smith et al. (2022). In their study, the authors define cold permafrost as permafrost with temperatures below -2°C. To provide additional clarity, we have included further explanations of this terminology in Section 4.1 and the caption of Figure 6.

L350 – Fig. 6 – It would be useful to give the number of sites considered in each region.

Author Response: The number of sites considered in the Andes was added to Figure 6 and Figure 10. The number of sites documenting long-term warming rates from other permafrost regions (Arctic, Sub-Arctic and Swiss Alps, aster Smith et al., 2022) were also added to Figure 6.

L354 – ALT was defined earlier so the sentence can be revised to "ALT in cryotic…". You might want to consult Riseborough (2008, ICOP) regarding using interpolation/extrapolation to determine ALT and the best approach to use.

Author Response: Thank you for the comment. We understand Riseborough's (2008, ICOP) points on extrapolating the zero-degree isotherm to estimate the active layer thickness (ALT) and agree

that this may be a more optimal approach. However, we lack sufficient measurements in the active layer to confidently extrapolate temperature data from above. Additionally, snow cover varies greatly at our sites. In areas with little snow, the active layer is not thermally insulated, and atmospheric gradients affect its temperature. In contrast, where snow is present, the permafrost table is influenced by the snow cover (e.g., BTS method by Haeberli, 1978).

Our main goal is to track potential changes and compare sites, which is best achieved using a consistent approach. Thus, we focus on relative comparisons rather than absolute ALT values.

We also acknowledge linear interpolation of the zero-degree isotherm may overestimate the active layer thickness, and, due to limited shallow data and variable snow cover, it should be considered a proxy, not the exact permafrost table depth.

We have adjusted the text to convey these ideas.

L369-370 – revision suggested "…overall increase in ALT over time…"

Author Response: Thank you for the comment. This section was edited for conciseness and to limit reference to long-term trends. The text now reads:

> *"There is no consistent increase in ALT over time across the dataset, and in some locations, the active layer may be shallowing."*

L369-372 – These results reflect short-term fluctuations in climate.

Author Response: Thank you for the comment. We have added text at the end of the sentence to indicate that these results reflect snort-term fluctuations in climate.

L373 – What is meant by deepening of permafrost? Referring to permafrost table or base? Be clear here.

Author Response: We are referring to the permafrost table, or top of permafrost. We have adapted the text to clarify, which now reads:

> *"In contrast, the top of permafrost is deepening in rock glaciers, at rates of approximately 0.4 m/yr at borehole 6-1, 0.8 m/yr at borehole 6-2 and 1.5 m/yr at borehole 6-4."*

L385 – Fig. 8. Is it possible to clearly delineate talik where it occurs so that it is clear to reader?

Author Response: Added labels to Figure 8 to indicate location of taliks.

L421 – Delete ""in this dataset"

Author Response: This text was adapted as part of editorial revisions to this paragraph. The sentence now reads:

*"Potential exceptions to this are noted in rock glaciers at Site 6 (i.e., boreholes 6-2, 6-5, 6-8 and 6-11), which, as noted above, are characterized by shallow isothermal conditions near ~ 0°C then increasing temperatures with depth."*

L436 Fig. 9 – some symbols are difficult to see

Author Response: Thank you for the comment. We have increased the size of symbols on the figure.

L439 – I think this section is good and makes use of the deeper temperatures that are available to give an indication of changes over several decades depending on depth.

Author Response: Thank you for the comment. No adjustments made.

L482 – suggested revision "..decrease as elevation increases (Figure...."

Author Response: Thank you for the comment. The text now reads:

*"Consistent with cooler temperatures at higher altitudes, both depth to permafrost and permafrost thickness decrease as elevation increases."*

L487 – Be clear if permafrost depth refers to permafrost table rather than base.

Author Response: We have adapted the text to clarify this point. The sentence now reads:

*"Excluding rock glaciers, the depth to top of permafrost decreases by approximately 1.9 m/km elevation gain"*

L488 – suggested revision "..increase as altitude increases at..."

Author Response: Thank you for the suggestion. We have revised this sentence to shorten, and the text now reads:

*"Excluding rock glaciers, the depth to top of permafrost decreases by approximately 1.9m/km elevation gain (r2 = 0.30), while permafrost thickness increases at approximately 300 m/km (r2 = 0.45)."*

L510-519 – This repeats information provided in the introduction and isn't necessary. You could start the section with line 520 and be clear that your study addresses a critical knowledge gap and then describe insights revealed etc.

Author response: Thank you for the comment. The first paragraph of S 5.1. was removed since the information was stated earlier. The following text was augmented to state that the study closes a critical knowledge gap in permafrost research in the region.

L525-526 – revision suggested "..allows for characterization of average ground temperature below seasonal influences and thermal gradients…."

Author Response: Thank you for the suggestion. We have adapted the text, which now reads:

*"Adequate depth of monitoring throughout the dataset enables characterization of average temperatures below the DZAA"*

L534 – revision suggested "…without adequate ground temperature data…"

Author Response: Thank you for the suggestion, the text now reads:

*"Another important implication of this work is that the data can be used to validate existing permafrost distribution models in the region, which were previously developed without any borehole temperature data"*

L562 – revision suggested "…of our sites in the Andes…."

Author Response: Thank you for the suggestion. The text now reads:

*"At several of our boreholes in the Andes"*

L572-573 – Lewkowicz et al (2012) is also relevant here – uses ground surface temperature and ground temperature in the analysis to show the colder ground conditions in valley bottoms.

Author Response: Included Lewkowicz et al (2012) as a reference to the concept of temperature inversions.

L628 – You can make a stronger statement here by replacing "illustrated" with "confirmed"

Author Response: Replaced "Illustrated" with "confirmed"

L632 – "from other boreholes" can be deleted (or replace with "from other sites")

Author Response: Deleted "from other boreholes"

L687 – (EIA) can be deleted since you give the term

Author Response: Thank you for the comment. We have deleted "(EIA)"

L693 – Delete "(GCOS)" since you give the term

Author response: Thank you for the comment. We have deleted "(GCOS)"

---

## Author Response (AR2)

Dear Dr. Bolch,

Thank you kindly for your review of our revised manuscript. Below is a point-by-point response to your comments from March 28, 2025, outlining the revisions we have made.

Comment 1:

L95 (of the track changes manuscript): Please do not cite so many references in a row, be more specific (as already mentioned in my access review). Here you can omit the references which are mentioned in the table as you refer to it.

Author Response:

Thank you for the suggestion. As recommended, we have removed the redundant citations in the text, as these references are already included in Table 1. The revised sentence now refers only to Table 1.

Comment 2:

L243: Is there no impact of freezing at all (also no seasonal freezing)? Please clarify.

Here you also kept the term "cryotic" and "non-cryotic" while is most other cases (except line 381) you changed to "permafrost". I am fine if you keep but just wanted to be sure that it is on purpose. If not than I suggest changing to be consistent.

Author Response:

The non-cryotic boreholes are indeed subject to seasonal freezing, and we have revised the sentence for clarity. It now reads:

> *"Of the total, 30 boreholes intercepted permafrost, while the remaining 23 were installed within unfrozen, or non-cryotic ground (Figure 2) subject only to seasonal freeze-thaw."*

Regarding the terminology, the use of "cryotic/non-cryotic" was intentional. Here, we provide the rationale for classifying boreholes with temperatures below 0 °C as "permafrost," even in the absence of two consecutive years of measurement. This is the point in the text where we explicitly justify the use of the term "permafrost" for these cases.

Comment 3:

L643f: Do not cite so many papers in a row; be more specific with the papers (e.g. mention the region they are covering).

Author Response: Thank you for the comment. We have removed the in-line citations and instead only reference Table 1, which lists the specific studies and their corresponding regions.

Comment 4:

Section 5.3 Shared Characteristics with other Mountain Permafrost Regions

While you discuss shared characteristics other regions almost no references of other regions are provided. As I did not provide a comment in this regard neither a reviewer did I do not want to ask at this stage, but still recommend to add some. Maybe you can also change the title as you put some emphasis on rock glaciers in this section.

Author Response:

Thank you for the suggestion. We have now added 2 relevant citations to support the discussion of shared characteristics with other mountain permafrost regions:

1- Etzelmüller, B., Guglielmin, M., Hauck, C., Hilbich, C., Hoelzle, M., Isaksen, K., Noetzli, J., Oliva, M., and Ramos, M.: Twenty years of European mountain permafrost dynamics—the PACE legacy, Environmental Research Letters, 15, 104070, https://doi.org/10.1088/1748-9326/abae9d, 2020.

And

2- PERMOS: Swiss Permafrost Bulletin 2022, edited by: Noetzli, J. and Pellet, C., 23 pp., 2023.

Regarding the title of Section 5.3, we have decided to keep it as is, as most of the boreholes discussed are not located in rock glaciers, and we do not wish to overemphasize these landforms within the dataset.

Thank you once again for your invaluable feedback. We have updated our documents with the changes made as 'technical corrections.' We greatly appreciate your input and hope that these revisions meet your expectations.

Best Regards,

Cassandra Koenig